# An alternative cloud index for estimating downwelling surface solar irradiance from various satellite imagers in the framework of a Heliosat-V method

Benoît Tournadre[1], Benoît Gschwind[1], Yves-Marie Saint-Drenan[1], Xuemei Chen[1], Rodrigo Amaro E Silva[1], and Philippe Blanc[1]

[1]MINES ParisTech, PSL Research University, O.I.E. - Centre Observation, Impacts, Energy, 06904 Sophia Antipolis, France

**Correspondence:** Philippe Blanc (philippe.blanc@mines-paristech.fr)

**Abstract.** We develop a new way to retrieve the cloud index from a large variety of satellite instruments sensitive to reflected solar radiation, embedded on geostationary as non geostationary platforms. The cloud index is a widely used proxy for the effective cloud transmissivity, also called clear-sky index. This study is in the framework of the development of the Heliosat-V method for estimating downwelling solar irradiance at the surface of the Earth (DSSI) from satellite imagery. To reach its versatility, the method uses simulations from a fast radiative transfer model to estimate overcast (cloudy) and clear-sky (cloud-free) satellite scenes of the Earth's reflectances. Simulations consider the anisotropy of the reflectances caused by both surface and atmosphere, and are adapted to the spectral sensitivity of the sensor. The anisotropy of ground reflectances is described by a bidirectional reflectance distribution function model and external satellite-derived data. An implementation of the method is applied to the visible imagery from a Meteosat Second Generation satellite, for 11 locations where high quality in situ measurements of DSSI are available from the Baseline Surface Radiation Network. For 15-minute means of DSSI, results from our preliminary implementation of Heliosat-V and ground-based measurements show a bias of 20 W m$^{-2}$, a root-mean-square difference of 93 W m$^{-2}$, and a correlation coefficient of 0.948. The statistics, except for the bias, are similar to operational and corrected satellite-based data products HelioClim3 version 5 and CAMS Radiation Service.

## 1 Introduction

Downwelling surface solar irradiance (DSSI) is one of the Essential Climate Variables defined by the Global Climate Observing System (GCOS, 2016). It is the solar part of the downwelling irradiance at the surface of the Earth and on an horizontal unit surface. The solar irradiance is defined as the integration on the spectral interval 290-3000 nm, accordingly to WMO (2014). DSSI considers the irradiance coming from all directions of the hemisphere above the surface: the irradiance coming from the direction of the Sun, usually referred to as beam horizontal irradiance, plus a diffuse component due to scattering caused by the atmosphere (clouds, gases, aerosols) and reflection by the surface, usually referred to as diffuse horizontal irradiance.

The knowledge of DSSI variations in space and time is of primary importance for various fields such as Earth sciences, solar energy industries, agriculture, or some medical fields. To meet all these needs, an ideal information on DSSI would feature high spatio-temporal resolution, a coverage of the entire Earth's surface, and the longest time period possible. Long time series

of data are notably useful to identify statistics of long-term inter-annual to multi-decadal variability and possible trends, if bias and standard deviation of the error requirements are reached.

Different approaches already exist to produce such DSSI data. Sources of data mainly include ground pyranometric measurements (Driemel et al., 2018), numerical weather prediction (NWP) modeling (Gelaro et al., 2017; Hersbach et al., 2020), and satellite-based remote sensing (Sengupta et al., 2021). Satellite-based methods are an efficient and accurate way to produce kilometric and sub-hourly resolved multidecadal time series of DSSI. A more comprehensive review of pros and cons of different methods is notably described in Huang et al. (2019).

The imagery produced by satellite radiometers provides a unique perspective on DSSI. Upwelling radiances coming from each location on Earth are acquired several times per day by a wide set of satellite imagers. This can particularly be achieved thanks to imagers embedded on meteorological geostationary (GEOs) and polar orbiting satellites. Another approach exists since 2015, thanks to the Deep Space Climate Observatory (DSCOVR) satellite mission: its Lissajous orbit around the L1 Lagrangian point between the Earth and the Sun makes it possible to picture the whole sunlit hemisphere of the planet, with a single satellite radiometer (Marshak et al., 2018; Hao et al., 2020).

Imagery of the Earth produced by satellite sensors exists for about six decades, and led early to the development of methods for estimating DSSI (Tarpley, 1979). Today, the information from multi-channel satellite measurements offers the possibility to derive cloud physical properties and then compute cloud attenuation of the solar radiation with methods like the Fast All-sky Radiation Model For Solar Applications (FARMS) (Xie et al., 2016), Heliosat-4 (Qu et al., 2017), Zhang et al. (2018), or Hao et al. (2019). Such methods are especially advantageous for highly reflective regions, where clouds are difficult to discriminate from the ground. Nevertheless, they require information on more than one spectral channel, limiting their versatility in the choice of satellite sensor.

The use of radiative transfer models and look-up tables is also quite common in the field of satellite-based estimation of DSSI, but usually requires pre-existing informations on cloud properties or a cloud mask (*e.g.* ISCCP-F (Zhang, 2004), GEWEX-SRB (Pinker and Laszlo, 1992; Cox et al., 2017), CLARA (Mueller et al., 2009), Cloud_cci (Stengel et al., 2020; Stephens et al., 2001), SICCS (Greuell et al., 2013)).

Another group of methods, labeled as "cloud-index methods", are able to produce estimates of downwelling surface solar irradiance from the visible imagery of satellite radiometers without external knowledge on cloud physical and optical properties. This gives them potential to retrieve multi-decadal time series including from the imagery of oldest 2-channel sensors like the Meteosat Visible and Infrared Imager (MVIRI). Cloud-index methods emerged quite early, notably with the seminal work of Möser and Raschke (1983) and the first Heliosat method (Cano et al., 1986; Cano, 1982). The cloud index quantity derives from the radiances measured by spaceborne sensors, and relates them to the extinction of the DSSI caused by clouds. The greater the cloud index, the greater the extinction, and the smaller the DSSI. More precisely, the cloud index can be used as an empirical proxy for effective cloud transmissivity. The latter, also named "clear-sky index" within the scientific community of solar energy, is defined as the ratio of the all-sky surface irradiance to the clear-sky surface irradiance (Long and Turner, 2008; Beyer et al., 1996), *i.e.* the DSSI in cloud-free conditions.

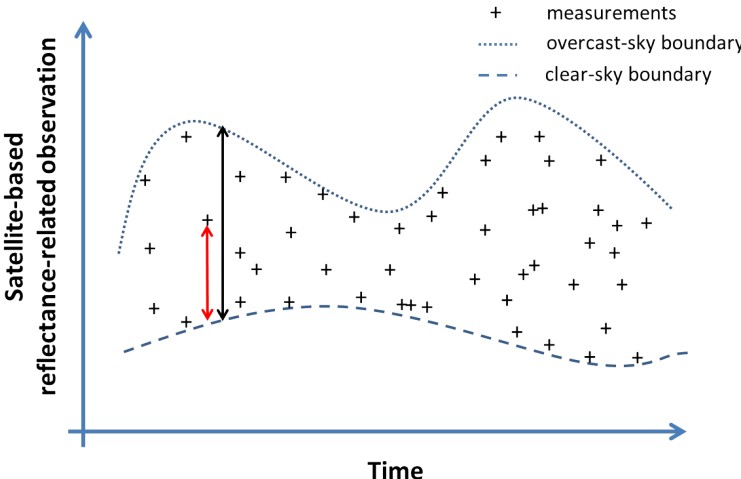

**Figure 1.** The calculation of a cloud index for a given location. The cloud index is the ratio between the distances "measurement to clear-sky" (red arrow) and "overcast-sky to clear-sky" (black arrow).

The cloud-index concept is based on the idea that the presence of a cloud brightens locally pixels of satellite shortwave imagery. In general, the value that quantifies reflectances of a given location observed from the top of the atmosphere (TOA), is comprised between a low and a high boundary values. The low boundary value $X_{\min}$ is taken as the clear-sky case and the high one $X_{\max}$ as the most cloudy case. The attenuation of downwelling surface solar irradiance by clouds is roughly given as a linear function of the difference between the measured value $X_{\mathrm{sat}}$ and the clear-sky boundary, relatively to the cloudy case - clear case difference, as illustrated in Figure 1. We name these variables $X$ as they can be of slightly different nature from one work to another (reflectance, albedo, radiance, digital count, etc.). The cloud index $n$ is then given as:

$$n = \frac{X_{\mathrm{sat}} - X_{\min}}{X_{\max} - X_{\min}} \tag{1}$$

Differences between cloud-index methods mainly rely on :

- modifications of the relationship between the cloud index and the effective cloud transmissivity (Zarzalejo et al., 2009; Perez et al., 2002; Gupta et al., 2001; Rigollier and Wald, 1998);

- the choice of the variable used to calculate the cloud index, for example TOA albedo (Darnell et al., 1988), reflectance (Wang et al., 2014; Gupta et al., 2001; Möser and Raschke, 1984)), Lambert equivalent reflectivity (LER) (Herman et al., 2018; Dave, 1978) or raw satellite numerical counts (Müller et al., 2015; Perez et al., 2002; Cano et al., 1986);

- the way to retrieve the $X_{\max}$ and $X_{\min}$ for the chosen variable.

Very different approaches are used to estimate the upper boundary, but for lower boundary, "archive-based" methods are used in most literature we reviewed: $X_{\min}$ is a minimum based on a time series of past satellite imagery. This approach

as some drawbacks. Firstly, it is hardly applicable to non-geostationary satellites due to variable viewing geometries and a low revisit time. As an example, Wang et al. (2014) use a climatology of surface albedo to derive DSSI from the Ozone Monitoring Instrument (OMI), embedded on the sun-synchronous satellite Aura. Secondly, systematic underestimations of the lower boundary $X_{\min}$ are commonly detected, for example due to dark shadowing caused by adjacent clouds on the surface,

aerosol treatment (Mueller et al., 2015). On the other hand, contamination of $X_{\min}$ by clouds on cloudiest regions can lead to systematic overestimation of $X_{\min}$. Finally, ensuring the observation of clear-sky instants by a sufficiently large time window and capturing the temporal variability of $X_{\min}$ by a sufficiently small time window is a difficult trade-off that can lead to biases if not well respected.

In this paper, we propose a cloud-index method based on radiative transfer modeling as an alternative to the archive-based

approach. This exploratory direction aims at reproducing the satellite measurements of reflectances in both clear-sky and overcast conditions based on description of surface, clear atmosphere and cloud properties. Radiative transfer simulations are able to reproduce how TOA reflectances depend on viewing and solar geometries, with also their spectral distribution. In addition, it is possible to provide to the radiative transfer model input data that describes variations in space and time of clear atmosphere composition and of surface properties. Thus, our approach is useful to identify and quantify sources of errors in

cloud-index methods.

With a spectral and angular description, our method is also able to extend the application field of the cloud-index approach to a wider variety of orbits and optical shortwave sensors. In order to limit the effects of molecular scattering, ozone absorption and polarization present in the ultraviolet, and of the absorption of radiation by clouds in the near infrared, the method focuses on satellite imagery in the spectral range 400-1000 nm ($\lambda < 1000$ nm). This range is wide enough to consider imagers on many

meteorological satellites launched since the beginnings of spaceborne Earth observation.

The method foreseen to compute the cloud index of Heliosat-V and eventually the DSSI is described in Section 2, along with the protocol of validation. Validation results are presented and discussed in Section 3 for simulated reflectances at the top of the atmosphere and for downwelling surface solar irradiance estimates. Section 4 is dedicated to the conclusion and perspectives.

## 2   Methods

Previous methods based on archives can avoid dependency on absolute calibration of the original imagery (Müller et al., 2015; Perez et al., 2002) and consider implicitely the anisotropy of $X_{\min}$. The pixel-to-pixel estimation of $X_{\min}$ is a surrogate for modeling the influence of viewing geometry on measured reflectances, while the slot-by-slot temporal characterization of $X_{\min}$ pictures the influence of varying solar-viewing geometry for the diurnal cycle of each pixel's reflectivity. The development of an alternative to archive-based approaches means dealing with new issues: a challenge is to reproduce explicitly and accurately

the TOA reflectances. For this, input data and models used need to satisfy the requirements for accurate DSSI estimations, as will be discussed therafter.

## 2.1 The cloud index $n$

As stated in the introduction, Heliosat-V is a method approximating the attenuation of DSSI radiation by clouds with a cloud index, $n$. Here, the cloud index components are reflectances considered at the top of the atmosphere (TOA), and corresponding to the satellite radiometer viewing geometry and spectral sensitivity. Reflectances are defined by the relation:

$$\rho = \frac{\pi\, L}{E_0\, \cos(\theta_{\mathrm{s}})} \tag{2}$$

with $L$ the upwelling radiance at TOA for a given spectral channel, $E_0$ the downwelling spectral solar irradiance at the top of the atmosphere on a perpendicular plan weighted by the spectral response function of the channel, and $\theta_{\mathrm{s}}$ the solar zenith angle for a given location (*i.e.* latitude and longitude) and a given time. $E_0$ varies mainly with the Sun-Earth distance, computed here with the Solar Geometry 2 algorithm (Blanc and Wald, 2012). The cloud index is then defined as:

$$n = \frac{\rho_{\mathrm{sat}} - \rho_{\mathrm{clear}}}{\rho_{\mathrm{ovc}} - \rho_{\mathrm{clear}}} \tag{3}$$

where $\rho_{\mathrm{sat}}$ is the reflectance measured by the radiometer for the given spectral channel, while $\rho_{\mathrm{clear}}$ and $\rho_{\mathrm{ovc}}$ are estimates of the reflectance that would be measured by the same sensor for, respectively, a clear-sky scene, and an overcast scene, *i.e.* with an optically thick cloud covering the whole pixel considered. The notion of "optically thick cloud" will be described in detail in the subsection 2.3.

Because of its definition, the cloud index may also be calculated with radiances. We consider here reflectances in order to visualize the anisotropic nature of different scenes. It has also the advantage to be a normalized quantity, so we can compare results for different radiometric channels and different SZAs.

The relationship between $n$ and DSSI varies slightly from one method to another, in particular for highest and lowest values of $n$. The core of the relationship for intermediate values of $n$ follows usually:

$$G = G_{\mathrm{c}}\, (1 - n) \tag{4}$$

where $G$ is the all-sky DSSI and $G_{\mathrm{c}}$ is the DSSI in clear-sky conditions and is provided by an external model. The external model used in this paper will be presented and discussed in section 2.4. The clear-sky index $K_{\mathrm{c}}$ is largely used to simplify the reading and is defined as:

$$K_{\mathrm{c}} = \frac{G}{G_{\mathrm{c}}} \tag{5}$$

so we can rewrite Equation (4) as:

$$G = G_{\mathrm{c}}\, K_{\mathrm{c}} \tag{6}$$

In this paper, we keep the original and simple relation $K_{\mathrm{c}} = 1 - n$ introduced by Darnell et al. (1988). Its improvement is out of the scope of this work but has been explored by various studies *e.g.* by Rigollier and Wald (1998) (reported in Rigollier et al. (2004)); Gupta et al. (2001); Perez et al. (2002); Zarzalejo et al. (2009), notably to better characterize cloudy situations with $n \approx 1$. In the following subsections, we describe the method used to compute $\rho_{\mathrm{clear}}$, $\rho_{\mathrm{ovc}}$ and $G_{\mathrm{c}}$.

## 2.2 The clear-sky reflectances $\rho_{\mathrm{clear}}$

We use a radiative transfer model to estimate what a spaceborne optical imaging system would measure in clear-sky conditions, for a given radiometric channel. Using simulations in cloud indices has previously been done notably to retrieve effective cloud fractions from the OMI instrument (Lorente et al., 2018; Veefkind et al., 2016; Stammes et al., 2008). We apply the same approach to satellite radiometers.

Radiative transfer simulations are able to estimate reflected radiation at the top of the atmosphere (TOA) considering the non-lambertian nature of the atmosphere and of Earth's surfaces. For the implementation of the method applied here, we use the model *uvspec* within the software package libRadtran (version 2.0.2) (Emde et al., 2016) and the one-dimension solver DISORT (Buras et al., 2011). We chose to use 32 streams for DISORT as a good compromise between time computation and a good angular representativeness of simulated radiances. For the spectral description, radiative transfer simulations are made following the so-called REPTRAN spectral approximation (Gasteiger et al., 2014). This parameterization enables the production of fast computations of radiative transfer adapted to the spectral sensitivity of satellite radiometric channels.

The spectral description of downwelling solar irradiance at the top of the atmosphere is provided by data from Kurucz (1992) for simulating $\rho_{\mathrm{clear}}$. The composition of the atmosphere is provided by time series of total atmospheric columns of ozone and water vapour, and partial aerosol optical depths (AOD) from the Monitoring Atmospheric Composition and Climate (MACC) reanalysis (Inness et al., 2013) distributed by the ECMWF. Data from MACC are extracted from the McClear service (http://www.soda-pro.com/web-services/radiation/cams-mcclear). MACC values are originally given on a 3-hour time step and with a spatial resolution of about 80 km (Inness et al., 2013; Lefèvre et al., 2013). The McClear service applies to MACC data a bilinear spatial interpolation onto the considered location, and a linear interpolation in time to a 1-min time step (Lefèvre et al., 2013). The atmospheric abundance profiles of $O_2$, $CO_2$ and $NO_2$ are kept to the fixed values of the Air Force Geophysics Laboratory (AFGL) midlatitude summer profile (Anderson et al., 1986), along with the temperature, pressure and air density profiles.

Partial AOD from MACC are provided at the wavelength 550 nm, for 5 types of aerosols (black carbon, dust, sea salt, organic matter, sulfate). Even though two supplementary classes "ammonium" and "nitrate" are now included in the Copernicus Atmospheric Monitoring Service (CAMS) reanalysis, these do not impact the method here proposed and were, thus, not considered.

An algorithm developed by Lefèvre et al. (2013) translates MACC partial aerosol optical depths information into aerosol mixtures designed for the Optical Properties of Aerosols and Clouds (OPAC) software package (Hess et al., 1998). These mixtures are associated to aerosol properties: scattering and absorbing coefficients, single scattering albedo, asymmetry parameter and the Angström coefficient. The total AOD at 550 nm is then calculated as the sum of partial AOD at 550 nm provided by CAMS. As libRadtran needs a total AOD input for the simulated wavelength, the OPAC Angström coefficient of the given mixture is used to estimate the AOD at the required wavelength.

An important component to simulate $\rho_{\mathrm{clear}}$ is the reflection properties of surfaces. The impact of the anisotropy of surface reflectance has notably been shown for estimates of a cloud index derived from measurements of the ultraviolet/visible Global

Ozone Monitoring Experiment 2 (GOME-2) and OMI by Lorente et al. (2018). The latter study also highlights the improvement of simulated shortwave clear-sky reflectances at the TOA, when using a model of bidirectional reflectance distribution function (BRDF) parameterized with data derived from the Moderate Resolution Imaging Spectroradiometer (MODIS) spaceborne instruments.

Here, we describe reflective properties of land surfaces with the RossThick-LiSparse (Ross-Li) model of bidirectional reflectance distribution function (Roujean et al., 1992; Lucht et al., 2000). It is then possible to consider the variations of the surface reflectance depending on viewing and solar zenith angles and on the azimuthal difference of both geometries $\Delta\phi$. The Ross-Li model decomposes the BRDF of a surface into a sum of three components: an isotropic contribution, independent of viewing and solar geometries; a volumic contribution, following a mathematical model of an idealized canopy ; and a geometric
contribution, considering the shadows induced by the roughness of the surface.

Algorithms have been developed to estimate the parameters $f_{\mathrm{iso}}$, $f_{\mathrm{vol}}$ and $f_{\mathrm{geo}}$ that weight respectively each of the contributors to the surface reflectance for all lands. This has notably been made with the imagery produced by the Moderate Resolution Imaging Spectroradiometer (MODIS) embedded on Terra and Aqua satellites (Wanner et al., 1997; Lyapustin et al., 2018).

We test here simulations with data from the Algorithm for Modeling MODIS Bidirectional Reflectance Anisotropies of the
185 Land Surface (AMBRALS) (Wanner et al., 1997) with its derived product MCD43C1 v6 (Schaaf et al., 2002). This product provides $f_{\mathrm{iso}}$, $f_{\mathrm{vol}}$ and $f_{\mathrm{geo}}$ parameters with a 0.05° resolution (about 6 km at the equator), a daily sampling rate, 16-day average and for seven spectral channels, including 4 channels in the 400-1000 nm spectral interval considered for the Heliosat-V method (Fig. 3). Owing to libRadtran documentation (Mayer et al., 2017), the values of each parameter are assigned to the central wavelength of its channel and a linear spectral interpolation is applied for the radiative transfer calculations. For
wavelengths shorter than the 0.47 $\mu$m MODIS channel, values are considered spectrally constant. For wavelengths longer than 0.85 $\mu$m, the interpolation is made between parameters at MODIS channels 0.86 $\mu$m and 1.24 $\mu$m.

## 2.3 The overcast-sky reflectances $\rho_{ovc}$

Cloud-index methods in the literature use various ways to estimate the TOA reflectances in overcast conditions $\rho_{\mathrm{ovc}}$ (Perez et al., 2002; Lefèvre et al., 2007; Mueller and Träger-Chatterjee, 2014). One way to approximate it without the use of archives
of satellite imagery has been proposed within the Heliosat-2 framework (Lefèvre et al., 2007) with an empirical relation based on the work of Taylor and Stowe (1984). It considers a dependency of $\rho_{\mathrm{ovc}}$ with the single solar zenith angle $\theta_{\mathrm{s}}$.

However, spectral radiative transfer simulations of $\rho_{\mathrm{ovc}}$ show that there is also a significant dependency between the TOA cloudy reflectances and other variables. In Figure 2, we represent 2-dimension histograms of TOA reflectances calculated from such simulations, with a solar zenith angle set to 30° as an example. For most wavelengths, a significant spread of
200 the distribution is observed (Fig. 2, two upper rows), corresponding only to different viewing geometries defined by a linear meshgrid in cosine of viewing zentih angle ($\theta_{\mathrm{v}}$) and difference $\Delta\phi$ of solar and viewing azimuth angles (Table 1).

In this paper, we assume a cloud optical thickness (COT) of 150 to define optically thick clouds and overcast conditions. This assumption relies on COT statistics from Trishchenko et al. (2001). The simulations for a low thick cloud (cloud top height (CTH) at 500 m) and a high thick cloud (CTH at 15 km) show in general a good agreement (Fig. 2, lower panel), except in

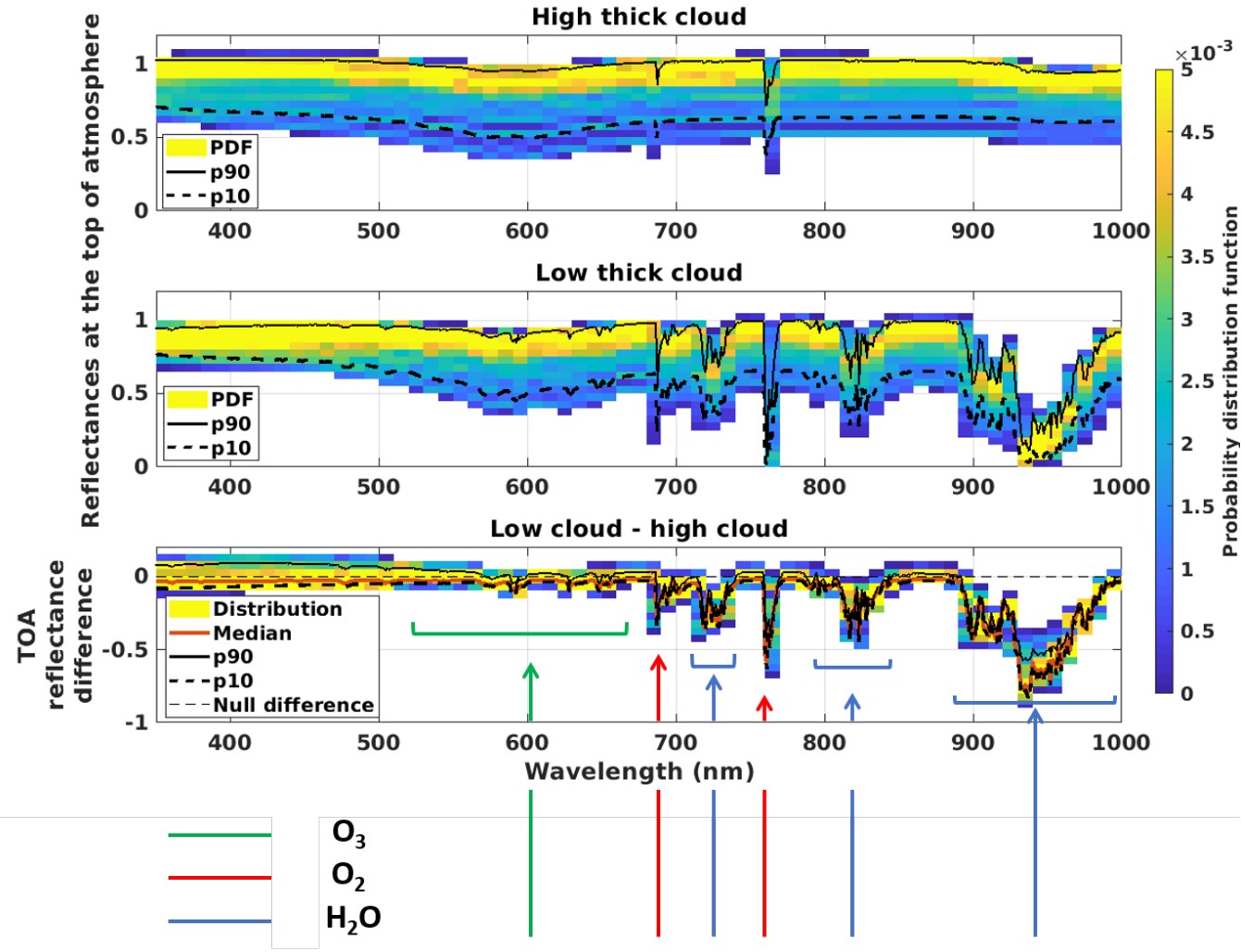

**Figure 2.** Two upper rows : distributions of simulated TOA reflectance spectra in overcast conditions $\rho_{ovc}$ for the different viewing geometries in the look-up table and for a solar zenith angle of 30°, with a thick liquid cloud (COT = 150). First row: CTH = 15 km ; cloud base height (CBH) = 2 km. Second row: CTH = 0.5 km ; CBH = 0.2 km . Third row: error on $\rho_{ovc}$ caused by a misattribution of cloud height to the "low thick cloud" category. Green, red and blue arrows indicate spectral regions with main absorption features from $O_3$, $O_2$ and $H_2O$, respectively.

absorbing bands of $O_2$ (mainly at 690 nm ($O_2$-B band) and 762 nm ($O_2$-A band)) and $H_2O$ (mainly at 725 nm, 820 nm and 950 nm) and for short wavelengths where scattering becomes increasingly significant (*e.g.* Jin et al. (2011)). For these wavelengths, the TOA reflectances with low clouds can be much lower than for high clouds, for a given cloud optical thickness. But outside these specific spectral regions, the height of clouds will not affect significantly the results of the method.

**Table 1.** Characteristics of the look-up table of cloudy TOA reflectances

| Characteristics | Values |
|---|---|
| Cloud phase | liquid (ice only for sensitivity tests) |
| Cloud optical thickness (COT) | 150 |
| Cloud droplet radius | vertical profile between 8 and 12 $\mu$m |
| Cloud top height (CTH) | 500 m ; 15 km |
| Cloud base height (CBH) | 200 m ; 2 km |
| Solar zenith angle ($\theta_s$) | 0° : 5° : 85° |
| cosine of viewing zenith angle ($\cos \theta_v$) | 0.1 : 0.1 : 1 |
| difference of solar and viewing azimuth angles ($\Delta\phi$) | 0°, 5°, 10° : 20° :170°, 175°, 180° |
| Spectral resolution | 1 nm |
| TOA spectrum | Gueymard (2018) |
| Ozone total column | 300 Dobson Units (DU) |
| Water vapour total column | 20 kg m$^{-2}$ |
| Aerosols | default aerosol described in Shettle (1990) |
| Temperature and pressure profiles | AFGL midlatitude summer |

An alternative way is therefore to produce look-up tables (LUT) from radiative transfer simulations, an approach notably applied in the framework of the HelioMont cloud-index method (Stöckli, 2014). It is then possible to take into account the viewing geometry and also the spectral variability of $\rho_{ovc}$. Assumptions have to be made on the properties of the optically thick clouds as the Heliosat-V method is designed to work by using only one spectral channel in the range 400-1000 nm: cloud top height, phase of cloud, cloud optical thickness, cloud droplet radius or ice crystal shape and size.

Here we construct a liquid cloud LUT of $\rho_{ovc}$, setting different cloud and atmosphere properties, geometry and spectral grids, as described in Table 1. The optical properties of the clouds come from the precalculated Mie tables provided by the libRadtran software package.

As no information is provided on the actual cloud vertical structure, $\rho_{ovc}$ are calculated as :

$$\rho_{ovc} = \frac{1}{2}(\rho_{ovc,high} + \rho_{ovc,low}) \tag{7}$$

where $\rho_{ovc,high}$ and $\rho_{ovc,low}$ are respectively derived from the high and low liquid cloud LUTs, interpolated on the viewing and solar geometries of the satellite time series and adapted to the spectral response function of radiometric channels.

An ice cloud LUT is also produced, to study the sensitivity of surface irradiance estimates to the assumed cloud phase. The ice cloud characteristics follow the parameterization by Yang et al. (2013). We use the "aggregate of 8 columns" ice crystal habit and the "severe" degree of roughness, which are notably used for the description of ice clouds in the look-up table of the MODIS collection 6 cloud product (Amarasinghe et al., 2017).

**Table 2.** Characteristics of the simulated reflectances at TOA in clear-sky conditions

| Characteristics | Values |
|---|---|
| Surface reflection model | RossThick-LiSparse |
| Surface reflection data | MODIS MCD43C1 v6 |
| Surface elevation | Shuttle Radar Topography Mission |
| Spectral resolution | *REPTRAN channel* parameterization (Gasteiger et al., 2014) |
| TOA spectrum | Kurucz (1992) |
| Ozone total column | ECMWF |
| Water vapour total column | ECMWF |
| Aerosol optical depth at 550 nm | sum of MACC partial optical depths |
| Aerosol mixtures and properties | Lefèvre et al. (2013) and OPAC |
| Temperature and pressure profiles | AFGL midlatitude summer |

## 2.4 The clear-sky model of surface irradiance $G_c$

The clear-sky surface irradiance is given by the the version 3 of the McClear model (Gschwind et al., 2019). The McClear model is a fast and accurate model that provides clear-sky estimation of DSSI with an absolute bias below 21 W m$^{-2}$ and a standard deviation error below 25 W m$^{-2}$ for six stations part of the reference Baseline Surface Radiation Network (BSRN) (Ohmura et al., 1998; Driemel et al., 2018), namely: Brasilia, Carpentras, Palaiseau, Payerne, Sede Boker and Tamanrasset. The McClear model was fed with the partial optical depths at 550 nm for black carbon, dust, sea salt, organic matter and sulfate from MACC reanalysis. It is also fed by water vapor atmospheric total columns and the ozone total columns provided by ECMWF. Data was dowloaded from the McClear web service (http://www.soda-pro.com/web-services/radiation/cams-mcclear).

## 2.5 Set-up and datasets for validation

The method has been tested on images from the Spinning Enhanced Visible and Infra-Red Imager (SEVIRI), aboard the Meteosat-9 meteorological geostationary satellite belonging to the family of Meteosat Second Generation (MSG). We consider measurements in the solar channels 0.6 $\mu$m and 0.8 $\mu$m channels, for the year 2011 and for 11 locations in the field of view of the satellite, corresponding to locations of pyranometric in situ sensors from the BSRN network. We use the calibration gains provided by EUMETSAT that operates MSG. For sensors with a linear count response like MSG/SEVIRI (Doelling et al., 2018), the radiance $L_{\text{sat}}$ is related to digital count $C$ via: $L_{\text{sat}} = g(C - C_0)$ where $C_0$ is the so-called space count.

To study the validity of the method, we compare DSSI estimates from MSG satellite measurements with pyranometric DSSI data retrieved from BSRN measurement stations. Considered stations are listed in Table 3 and displayed in the MSG field of view in Figure 4. Only the highest quality BSRN measurements of surface irradiance are used, having passed a quality check (Lefèvre et al., 2013). Figure 5 shows the time series when data are considered valid, for each station.

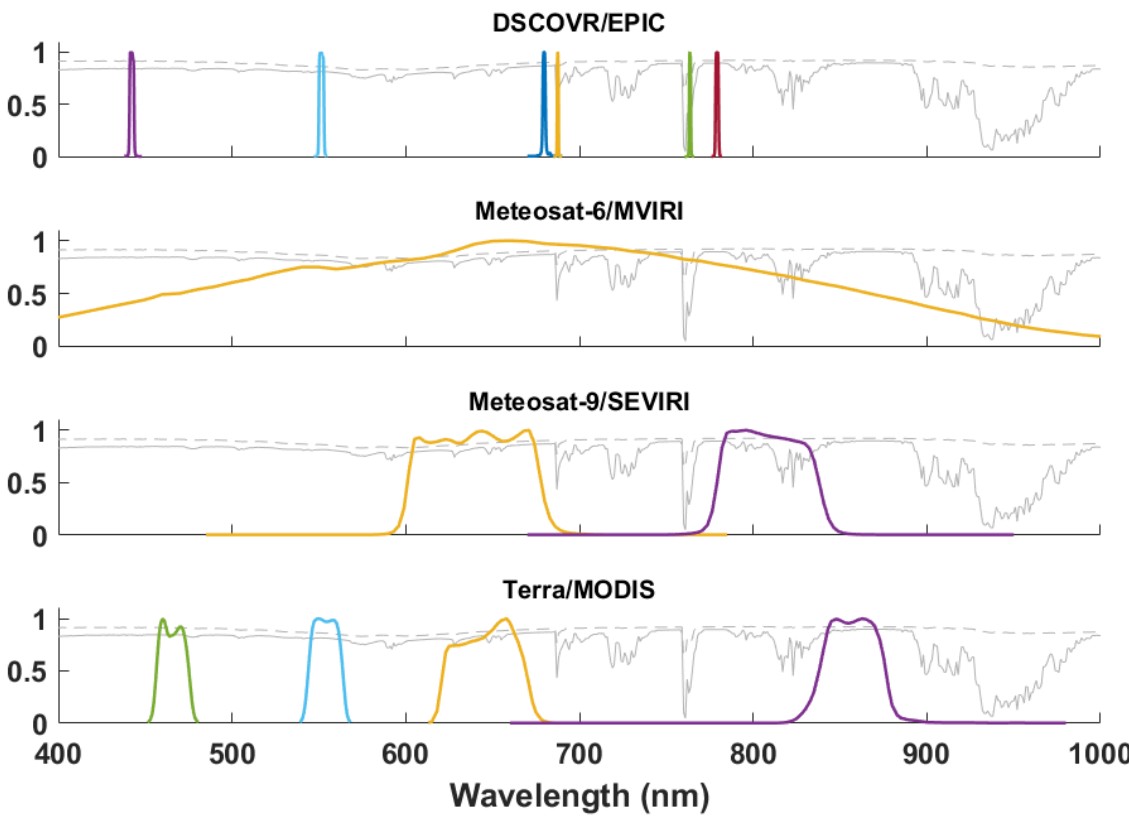

**Figure 3.** Colored lines: spectral response functions of different sensors in the spectral range considered by Heliosat-V. Gray lines : TOA reflectance spectra of typical scenes with a high (dashed line) and low altitude (solid line) thick cloud

**Table 3.** List of BSRN stations used for validation

| Station | Code | Latitude | Longitude | Elevation |
|---------|------|----------|-----------|-----------|
| Brasilia | BRB | 15.6010° S | 47.7130° W | 1023 m |
| Cabauw | CAB | 51.9711° N | 4.9267° E | 0 m |
| Camborne | CAM | 50.2167° N | 5.3167° W | 88 m |
| Carpentras | CAR | 44.083° N | 5.590° E | 100 m |
| CENER | CNR | 42.8160° N | 1.6010° W | 471 m |
| Lindenberg | LIN | 52.2100° N | 14.1220° E | 125 m |
| Palaiseau | PAL | 48.7130° N | 2.2080° E | 156 m |
| Payerne | PAY | 46.8150° N | 6.9440° E | 491 m |
| Sede Boker | SBO | 30.8597° N | 34.7794° E | 500 m |
| Sao Martinho da Serra | SMS | 29.4428° S | 53.8231° W | 489 m |
| Tamanrasset | TAM | 22.7903° N | 5.5292° E | 1385 m |

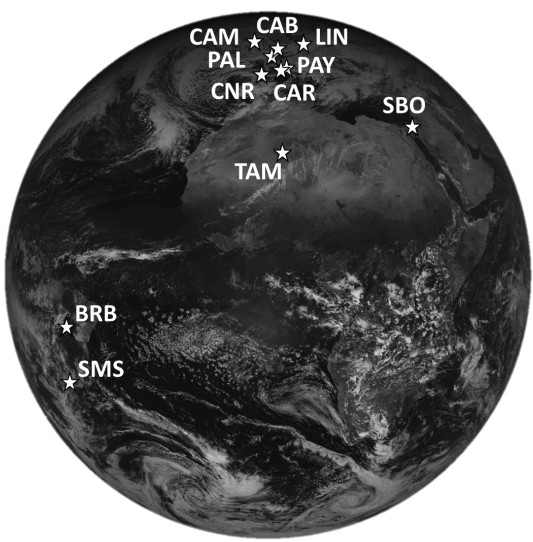

**Figure 4.** BSRN ground stations used for validation in this study, in the field of view of Meteosat Second Generation (0.6 $\mu$m channel).

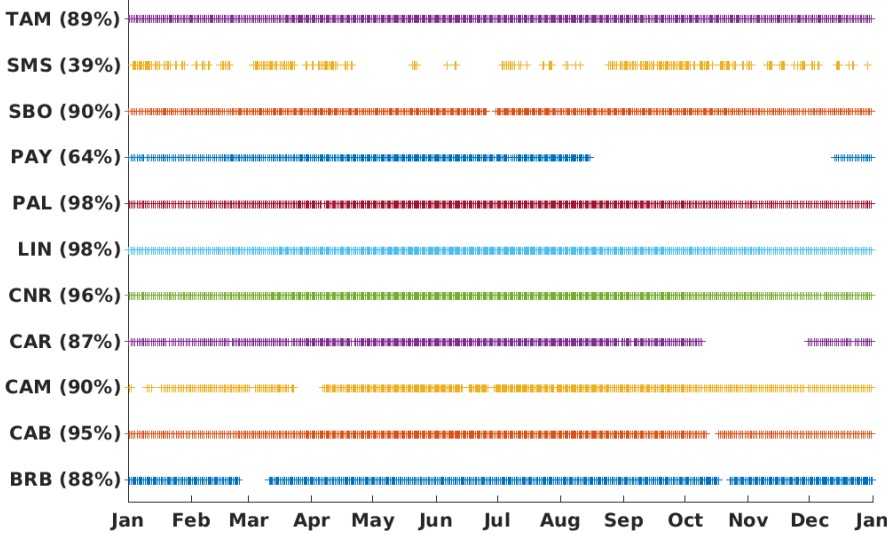

**Figure 5.** Time series used for the 15-min mean statistics between satellite estimates and quality-checked BSRN measurements during the year 2011. In parentheses : percentage of data conserved.

We also compare the results of our method to operational satellite-based products of surface irradiance. For this, we use
data from HelioClim3 version 5 (HC3v5) and CAMS Radiation (CAMS-RAD) DSSI databases. Both are derived from the
imagery of the SEVIRI sensor and are produced by a Heliosat method: a modified version of Heliosat-2 for HC3v5 (Qu
et al., 2014) and Heliosat-4 for CAMS-RAD. Both products and their descriptions are provided by the SoDa service (http:
//www.soda-pro.com/).

As this work is exploratory on a new method, we limit ourselves to conservative situations with solar zenith angles lower
than 80°, covering most cases. For higher angles, some effects not considered by the method can occur, including shadowing
and high parallax effects.

## 3   Results and discussions

### 3.1   Validity of cloud index components

The validity of cloud index components, $\rho_{\mathrm{sat}}$, $\rho_{\mathrm{clear}}$, and $\rho_{\mathrm{ovc}}$, defines the uncertainty of $n$. From Equation (3), the uncertainty
on the cloud index can be written as:

$$\delta n = \left( \frac{\partial n}{\partial \rho_{\mathrm{sat}}} \right) \delta \rho_{\mathrm{sat}} + \left( \frac{\partial n}{\partial \rho_{\mathrm{clear}}} \right) \delta \rho_{\mathrm{clear}} + \left( \frac{\partial n}{\partial \rho_{\mathrm{ovc}}} \right) \delta \rho_{\mathrm{ovc}} \tag{8}$$

This leads to

$$\delta n = \frac{1}{\Delta} \left( \delta \rho_{\mathrm{sat}} - (1-n) \, \delta \rho_{\mathrm{clear}} - n \, \delta \rho_{\mathrm{ovc}} \right) \tag{9}$$

where $\Delta = \rho_{\mathrm{ovc}} - \rho_{\mathrm{clear}}$. It appears that the "clear-sky error" $(1-n) \, \delta \rho_{\mathrm{clear}}$ will be more significant in clear-sky conditions
(i.e., $n$ is close to 0), and the "overcast-sky error" $n \, \delta \rho_{\mathrm{ovc}}$ will be more important in overcast conditions (i.e., $n$ is close to
1). Besides, the error on cloud index will be inversely proportional to $\Delta$, the difference between overcast and clear-sky TOA
reflectances. Because of this relationship between the errors on cloud index and reflectances, the discussions in this section are
focused on absolute values of reflectance errors.

### 3.1.1   Measured reflectances at the top of the atmosphere

A potential important source to the measurement error $\delta \rho_{\mathrm{sat}}$ comes from the calibration gain. The operational calibration gains,
that we use in this paper, have a claimed uncertainty of around 4% (EUMETSAT, 2019). On the other hand, Hewison et al.
(2020) assert that the alternative method by Doelling et al. (2018), used for GSICS corrected computation of calibration gain,
limits its bias to below 1% .

The use of optimal calibration is out of the scope of our work. Still, we compared gain coefficients proposed by EUMETSAT
$g_{\mathrm{EUM}}$ with those provided by Doelling et al. (2018) $g_{\mathrm{D2018}}$ for the measurements produced by the Meteosat-9 0.6 and 0.8 $\mu$m
channels in 2011. They show a mean relative disagreement, calculated as $(g_{\mathrm{EUM}} - g_{\mathrm{D2018}})/g_{\mathrm{D2018}}$, of about -9 % for 0.6 $\mu$m
and -8 % for 0.8 $\mu$m during this period (also illustrated on Fig. A1). We expect that these errors will affect with the same

magnitude the agreement between numerical simulations and measurements of clear-sky TOA reflectances. This underlines that an accurate source of absolute calibration is important for the Heliosat-V method.

### 3.1.2 Simulated reflectances at the top of the atmosphere in clear-sky conditions $\rho_{\text{clear}}$

As an intermediate assessment, simulated clear-sky reflectances at the top of the atmosphere (TOA) $\rho_{\text{clear}}$ are compared to satellite measurements. Cloudy instants are manually filtered out of the satellite time series. Results comprising all the manually-filtered clear-sky instants in 2011 for all the eleven sites, are shown in Figure 6 as 2D reflectance histograms.

For the 0.6 $\mu$m and 0.8 $\mu$m channels, correlation coefficients are both higher than 0.9, but the correlation is much better for 0.6 $\mu$m with a value of 0.974. This means that the variability of $\rho_{\text{clear}}$ is significantly better represented, with almost 95 % of the total variance, for 0.6 $\mu$m than for 0.8 $\mu$m, with 82 % of the total variance. The root-mean-square difference (RMSD) between the simulated reflectance and measured reflectance in the clear-sky conditions is 0.03 (15%) for the 0.6 $\mu$m channel and 0.04 (12%) for the 0.8 $\mu$m channel. The bias is 0.02 (10%) and -0.02 (-7%) for 0.6 $\mu$m and 0.8 $\mu$m channels respectively, contributing a big part to the RMSD. The standard deviation of the difference (STD) is 0.02 for the 0.6 $\mu$m channel and 0.04 for the 0.8 $\mu$m channel. Both higher bias and STD for the 0.8 $\mu$m will contribute to lower the precision in the calculation of the cloud index based on this channel, compared to 0.6 $\mu$m. When studying station by station, the highest absolute standard deviation of the difference between simulations and measurements is reached for Sede Boker with 0.03, while the lowest is reached for Tamanrasset with 0.008. For bias, highest mean values reach +0.035 for 0.6 $\mu$m (Payerne) and -0.07 for 0.8 $\mu$m (Camborne) (see also Fig. B1). Using the gain coefficients developed by Doelling et al. (2018) for CERES-SYN1deg instead of EUMET-SAT operational coefficients is sufficient to remove most of the mean bias observed between simulations and measurements of $\rho_{\text{clear}}$, for the channel 0.6 $\mu$m. Besides, it increases the mean bias for the 0.8 $\mu$m channel.

It is worth noting that we use MCD43C1v6 BRDF data regardless of their quality flags. We observe though that keeping only the highest quality data improves significantly statistics (Figure B2). Also, the choice of a spectral linear interpolation between MODIS channels to simulate surface reflectances in SEVIRI channels is supposed to contribute significantly to biases observed in $\rho_{\text{clear}}$ simulations, in particular for the 0.8 $\mu$m channel with vegetated surfaces due to the red edge spectral pattern (low reflectivity below around 700 nm, high reflectivity above around 750 nm). Another part of the bias, difficult to quantify, is linked to the accuracy of the calibration of satellite measurements.

Figure 7 shows the diurnal variations of measured and simulated reflectances for the SMS and CAM stations. Both SMS and CAM are surrounded mainly by various types of vegetation and some urban area for the case of CAM (Figure B3). We observe that simulations are able to reproduce partly the diurnal variability observed in clear-sky conditions (also refer to Figure B4 for channel 0.6 $\mu$m and 0.8 $\mu$m under different surface conditions). On Figure 8, we compare $\rho_{\text{clear}}$ values with the surface reflectance $\rho_{\text{surface}}$, computed with the RossThick-LiSparse model applied to BRDF parameters derived from MODIS 646 nm channel, and using viewing and solar geometries considered. Firstly, we see that $\rho_{\text{clear}}$ values are significantly higher than $\rho_{\text{surface}}$ with a different diurnal pattern. This shows the importance of considering the atmosphere anisotropic reflectance to reproduce TOA reflectances. We also can see the contribution from the surface anisotropy in the $\rho_{\text{clear}}$ simulations. This

appears in particular close to the backscattering direction where surface reflectance is enhanced: around noon in Camborne and the morning in São Martinho da Serra.

For CAM, some higher values of $\rho_{\text{clear}}$ are observed in January. This can be attributed to high aerosol optical depth during this period, as illustrated in Figure 9. It shows that $\rho_{\text{clear}}$ is not only sensitive to time variations of surface properties but also to atmospheric composition changes.

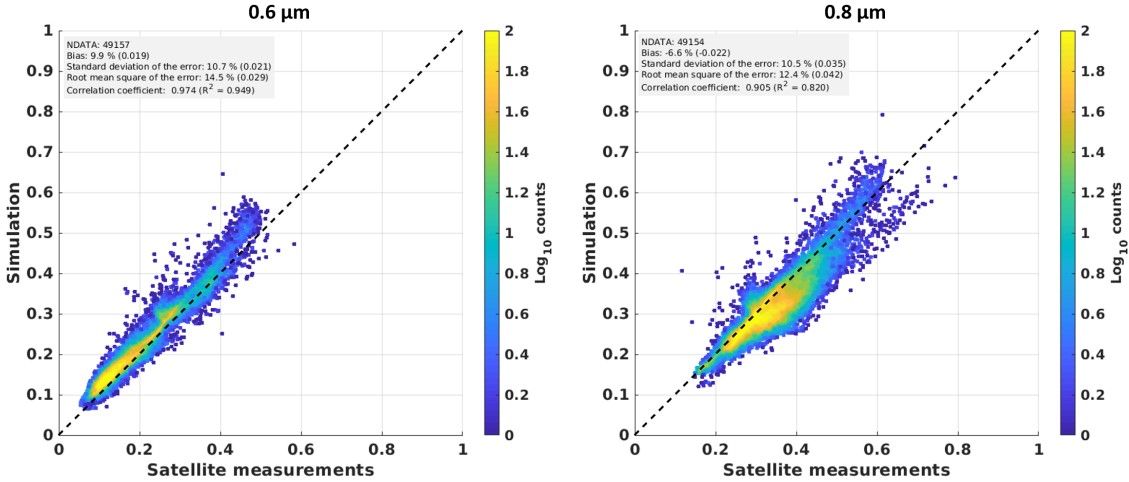

**Figure 6.** Simulation of clear-sky reflectances at the TOA ($\rho_{\text{clear}}$) for MSG 0.6 $\mu$m (left panel) and 0.8 $\mu$m (right panel) spectral channels compared with actual satellite measurements. Represented data include simulations and measurements for all 11 locations, for the year 2011.

### 3.1.3 Simulated reflectances at the top of the atmosphere in overcast conditions $\rho_{\text{ovc}}$

The validity of $\rho_{\text{ovc}}$ is more difficult to test than that of $\rho_{\text{clear}}$ by comparing with satellite measurements as the occurence of optically thick clouds can be rare depending on the location, the season and the hour of the day. We therefore use 9 years of Meteosat measurements, between 2011 and 2019 to extract the 1% most reflective scenes for each station, month and hour of the day (orange dots on Fig. 7). On the first row of Figure 7, we can see that some patterns are similar in simulated $\rho_{\text{ovc}}$ and 99th percentile of measurements over the São Martinho da Serra pixel: in the forward scattering conditions (evening on the West edge of Meteosat disc), both agree on increased values of $\rho_{\text{ovc}}$. On the other hand, some stations show regularly values of measured reflectances beyond the $\rho_{\text{ovc}}$ simulated boundary, as in the example of Camborne (Fig. 7, second row). Figure 7 illustrates also how $\rho_{\text{ovc}}$ depends on the liquid or ice phase of the cloud, due to their different scattering phase functions. Our ability to reproduce reflectances at the top of the atmosphere in overcast conditions depends therefore on our knowledge of cloud properties, including their scattering phase function and top height. Other effects like the tridimensional structure of

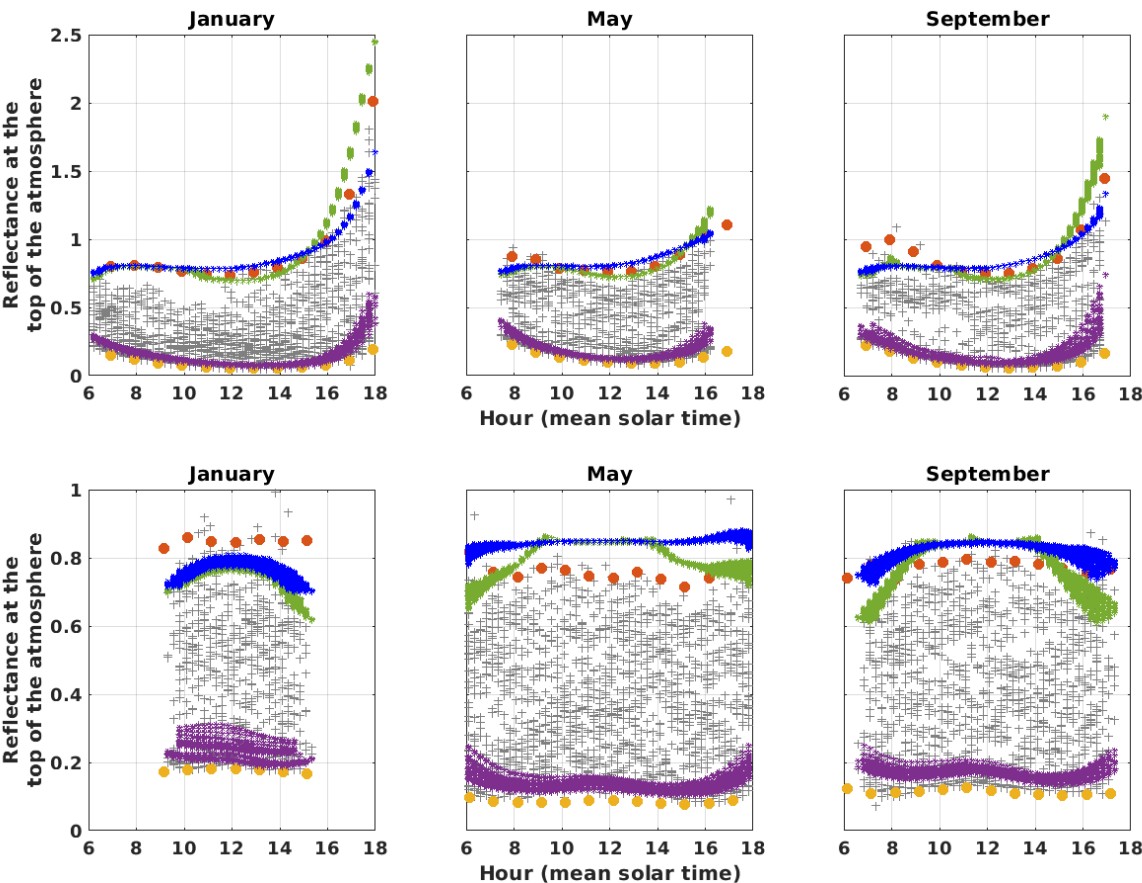

**Figure 7.** Simulated and measured reflectances at the top of the atmosphere above São Martinho da Serra (Brazil, upper row) and Camborne (United Kingdom, lower row) locations, for MSG 0.6 $\mu$m channel and for January, May and September calendar months. Grey plus signs: MSG measurements (2011, Meteosat-9). Green asterisks: reflectances in overcast conditions $\rho_{\rm ovc}$, derived from the liquid-cloud look-up table. Blue asterisks: same from the ice-cloud look-up table. Purple asterisks: reflectances in clear-sky conditions $\rho_{\rm clear}$, derived from radiative transfer simulations. Yellow and orange dots are respectively hourly percentiles 1 and 99 of MSG satellite measurements from year 2011 to 2019.

clouds probably explain part of the discrepancies between measurements and plane-parallel simulations in overcast conditions (Horvath and Davies, 2004).

### 3.1.4  Difference between simulated overcast and clear-sky reflectances

The difference $\Delta$ between overcast and clear-sky reflectances is bigger when the overcast reflectance is relatively low and clear-sky reflectance is relatively high. High values of $\Delta$ mean a good quality of cloud index estimation (cf. Equation 9). We study the dependencies of $\Delta$ with the simulated reflectances to identify conditions that will cause high uncertainties on the

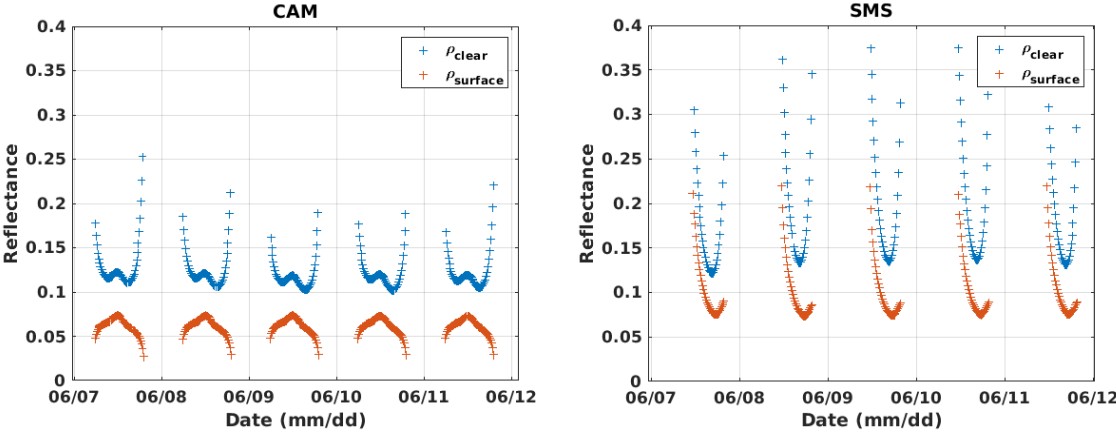

**Figure 8.** Comparison between simulations of clear-sky reflectances at the top of the atmosphere for MSG 0.6 $\mu$m channel ($\rho_{\mathrm{clear}}$, blue plus signs) and corresponding surface reflectances computed with the RossThick-LiSparse model applied to MODIS MCD43C1v6 BRDF parameters for the channel 646 nm ($\rho_{\mathrm{surface}}$, red plus signs) for five days in June 2011. Left panel: Camborne station (CAM) ; right panel: São Martinho da Serra station (SMS).

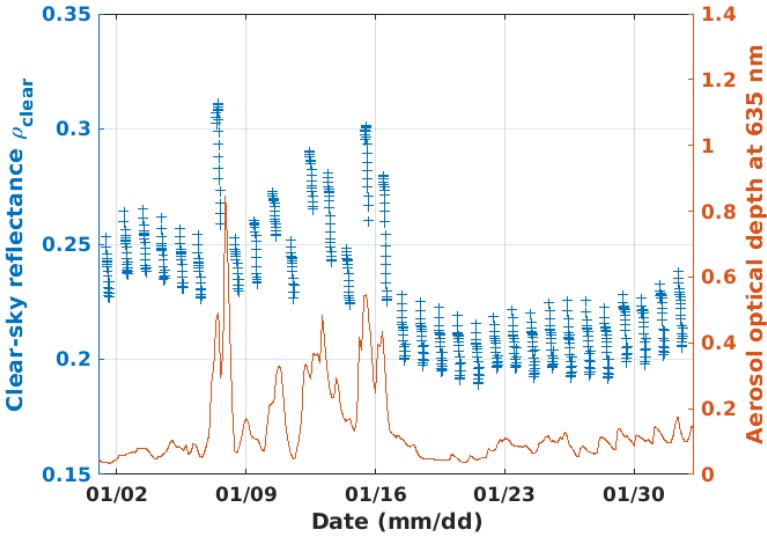

**Figure 9.** Blue plus signs: simulated reflectances at the top of the atmosphere in clear-sky conditions $\rho_{\mathrm{clear}}$ in January 2011 at Camborne station (CAM) and for MSG 0.6 $\mu$m. Red line: aerosol optical depth at 635 nm used for simulations.

computation of the cloud index. In general, we observe that the computed value of $\Delta$ is higher for the 0.6 $\mu$m channel than for 0.8 $\mu$m, as a combination of surface, cloud and clear atmosphere spectral signatures. This is illustrated in Figure 10 for

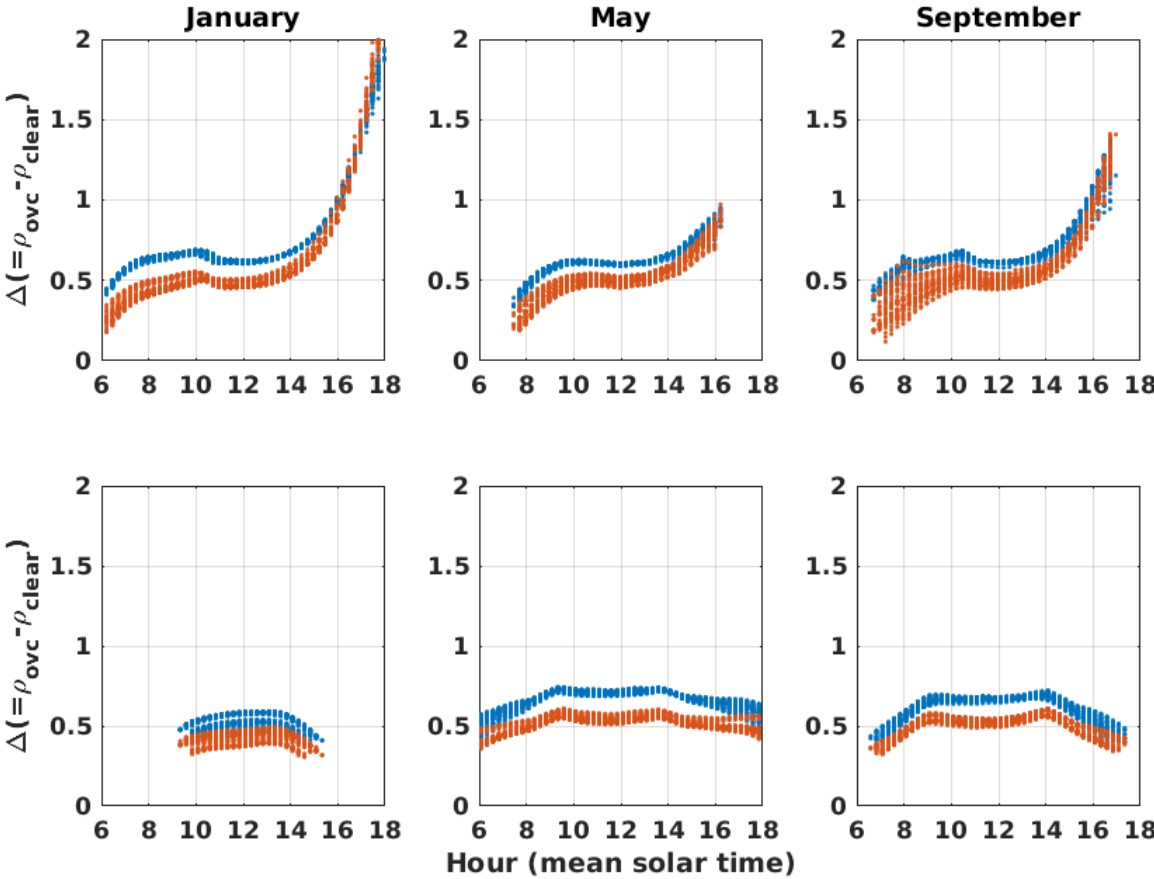

**Figure 10.** Difference between simulated reflectances at the top of the atmosphere in overcast and in clear-sky conditions $\Delta = \rho_{\mathrm{ovc}} - \rho_{\mathrm{clear}}$ for São Martinho da Serra (Brazil, upper row) and Camborne (United Kingdom, lower row) locations and for January, May and September calendar months (three columns from left to right). In blue dots: MSG 0.6 $\mu$m channel ; in red dots: MSG 0.8 $\mu$m channel.

stations SMS and CAM. We observe however for the desert stations TAM and SBO that both channels present similar values of $\Delta$ (Fig. B5). $\Delta$ depends also on the viewing and solar geometries because of $\rho_{\mathrm{ovc}}$ and $\rho_{\mathrm{clear}}$ different angular signatures. It leads for example for SMS station and channel 0.8 $\mu$m to low values of $\Delta$ in January morning and high values of $\Delta$ in the evening, which can be explained by the strong forward scattering of clouds occuring in these conditions.

### 3.2 Comparison of satellite-based estimates of DSSI with ground-based measurements

Validation results are shown in Table 4, for 15-min averaged DSSI estimates. Satellite-based estimates are obtained with MSG 0.6 $\mu$m imagery. Results for MSG 0.8 $\mu$m imagery show generally lower quality in terms of correlation and STD as shown in Figure 11.

The simple relationship between the cloud index and the clear-sky index used here explains the significant amount of negative values of DSSI estimates. The improvement of this relation will be the object of a future study.

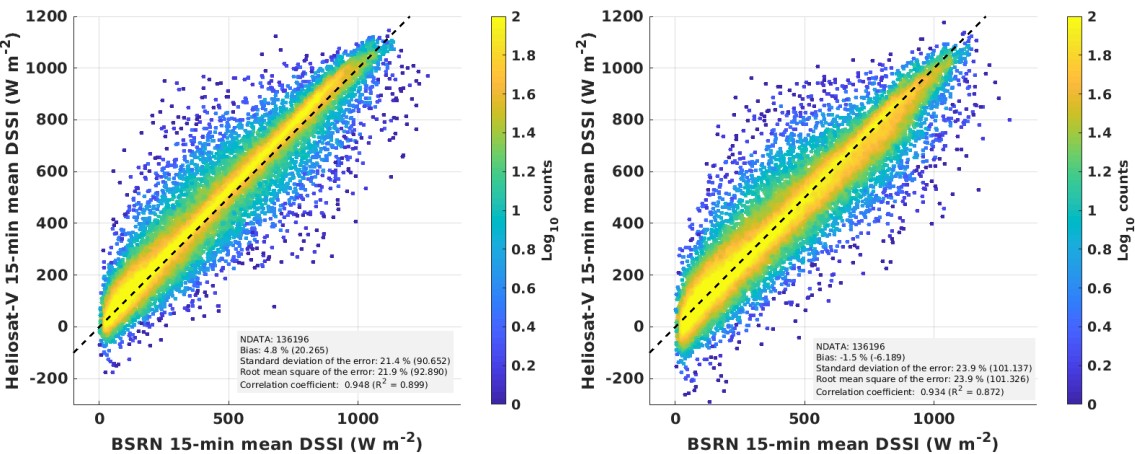

**Figure 11.** 2D-histograms of satellite-based DSSI estimates from the Heliosat-V method versus ground-based BSRN measurements for MSG 0.6 $\mu$m channel (left panel) and 0.8 $\mu$m channel (right panel).

We tested the sensitivity of the DSSI estimates to the cloud phase by using in one case the reference look-up table, featuring a liquid cloud, and for the test case, an ice cloud as described in Section 2.3. Results show only minor differences, pointing out a limited influence of the cloud phase on DSSI estimates (Fig. B6).

Finally, the quality of the results depends also on the quality of the clear-sky surface irradiance model. Gschwind et al. (2019) report for example relative mean biases of the McClear model from -3.6% (Barrow, Alaska, USA) to +3.2% (Payerne,
Switzerland), when compared to BSRN irradiance measurements. The improvement towards a least biased estimation of the downwelling surface solar irradiance based on a cloud index will require better estimates of the attenuation of the solar radiation by the clear atmosphere.

**Table 4.** Validation results for 15-min means of all-sky DSSI, for the year 2011. Results based on the imagery of Meteosat-9/SEVIRI 0.6 $\mu$m channel.

| Station | Number of samples | Mean DSSI (BSRN) W m$^{-2}$ | Bias W m$^{-2}$ (%) | RMSD W m$^{-2}$ (%) | Correlation coefficient (R) |
|---|---|---|---|---|---|
| Brasilia | 13570 | 504 | 25 (5) | 137 (27) | 0.883 |
| Cabauw | 13222 | 301 | 4 (1) | 72 (24) | 0.949 |
| Camborne | 12731 | 310 | -3 (-1) | 103 (33) | 0.901 |
| Carpentras | 12642 | 452 | 41 (9) | 80 (18) | 0.969 |
| CENER | 14164 | 412 | 21 (5) | 89 (22) | 0.946 |
| Lindenberg | 13637 | 317 | 9 (3) | 81 (26) | 0.938 |
| Palaiseau | 13993 | 335 | 12 (4) | 79 (24) | 0.948 |
| Payerne | 9191 | 387 | 29 (8) | 88 (23) | 0.955 |
| Sede Boker | 13574 | 589 | 46 (8) | 90 (15) | 0.960 |
| Sao Martinho da Serra | 5864 | 501 | 8 (2) | 102 (20) | 0.936 |
| Tamanrasset | 13609 | 579 | 26 (5) | 88 (15) | 0.958 |
| Total | 136197 | 436 | 20 (5) | 93 (22) | 0.948 |

### 3.3 Comparison of satellite-based estimates of DSSI with HelioClim3 and CAMS Radiation products

The results of the method are also compared to satellite-based DSSI products HelioClim3 version 5 (HC3v5) and CAMS Radiation Service (CAMS-RAD) on Table 5. Results for the new HSV method show statistics similar to HC3v5 and CAMS-RAD, for both estimates based on 0.6 $\mu$m and 0.8 $\mu$m channels, in terms of correlation and of STD. One may note very low values of bias for operational products. This is expected because CAMS-RAD and HC3v5 estimates are calibrated with DSSI measurements from a similar set of BSRN stations.

Better results from the channel 0.6 $\mu$m could be attributed to a smaller influence of the cloud top height, compared to the 0.8 $\mu$m channel which is affected by water vapour absorption (Fig. 3). Biases discussed for the computation of clear-ky and overcat TOA reflectances could also affect significantly DSSI estimates.

### 4 Conclusion and perspectives

Heliosat-V is a cloud-index method for estimating downwelling surface solar irradiance from satellite imagery. In the framework of its development, we proposed an alternative way to retrieve the components of cloud index, this index being used to quantify the attenuation of DSSI by clouds. The method takes advantage of radiative transfer modeling to provide versatility to the concept of cloud index. It provides advantages: it is applicable for optical sensors on geostationary and non-geostationary

**Table 5.** Comparison between validation results of HSV with those of HC3v5 and CAMS-RAD, each one versus BSRN measurements. Statistics on 15-minute means of DSSI for the stack of 11 stations and the year 2011. N = 135107 ; BSRN mean = 424 W m$^{-2}$

| Method/data product | Bias<br>W m$^{-2}$ (%) | STD<br>W m$^{-2}$ (%) | RMSD<br>W m$^{-2}$ (%) | Correlation coefficient<br>(R) |
|---|---|---|---|---|
| HSV 0.6 $\mu$m | 20 (5) | 91 (21) | 93 (22) | 0.948 |
| HSV 0.8 $\mu$m | -6 (-2) | 101 (24) | 101 (24) | 0.934 |
| HC3v5 | 2 (0) | 88 (21) | 88 (21) | 0.950 |
| CAMS-RAD | 0 (0) | 98 (23) | 98 (23) | 0.937 |

orbits, flexible for future improvements to describe surface, clear atmosphere and clouds and investigates physical solutions for limitations observed in previous cloud-index methods.

the method can be applied to different satellite optical sensors embedded on geostationary as non-geostationary orbits, it provides flexibility for future improvements to describe surface, clear atmosphere and clouds and solves some limitatio. Also, it provides an explicit An alternative cloud-index method is described in the framework of the development of the Heliosat-V method for estimating downwelling surface solar irradiance from satellite imagery. The proposed method uses a radiative transfer model to compute the theoretical lower and upper boundaries of satellite measurements, corresponding to the clear-sky and overcast reflectances at the top of the atmosphere. These simulations, along with the satellite measurements, are used to compute the cloud index needed to quantify the attenuation of DSSI by clouds. is built to deal with a single radiometric channel in the spectral range 400-1000 nm. It also does not need archives of data to quantify the cloud effective transmissivity. This approach has advantages. First, the concept of the Heliosat-V cloud index enables the use of imagery from geostationary and non-geostationary platforms, an asset to reach an extended spatial coverage. Moreover, the approach has the potential to deal with long time series of imagery from radiometers characterized by different spectral sensitivities and viewing geometries.

Validation results using SEVIRI imagery show that DSSI can be estimated by a cloud index method that does not rely on archives of imagery, with a quality similar to operational satellite-based data products like CAMS Radiation Service and HelioClim3, in terms of RMSD and correlation. This is an encouraging step toward the application of a Heliosat method to non geostationary satellite sensors. However, we note that there are differentiated errors depending on the spectral channel considered. This could be attenuated notably by an external knowledge on cloud top height and by improving the spectral interpolation of reflexion properties of vegetated surfaces.

To clarify the potential of the method for long time series of imagery, we will need to explore how sensitive to the quality of input data the results are. The knowledge on atmospheric composition in absorbing and scattering species and on surface reflectivity properties is notably lower for past periods like 1980's than for today. Also, the absolute calibration of satellite imagery can be more uncertain, without on-orbit calibrated instruments. Many inputs of the method have very different degrees of quality, depending on the period considered: the composition of the clear-sky atmosphere (aerosols and gases), surface

properties, external clear-sky irradiance model. Further work is still to be done on multidecadal time series to study how the quality of such ancillary data affect the estimates of DSSI.

Also, producing global maps of DSSI requires to deal with non geostationary satellite imagery. First tests of the method have been made on the imagery of the Earth Polychromatic Imaging Camera (EPIC) embedded on the DSCOVR platform. They

show encouraging results that will be extended and detailed in a future publication.

Global coverage of DSSI information obviously requires also to deal with ocean surfaces and snow covered regions, and this will need to be treated in the future.

*Code availability.* Excerpts of code are available at https://cloud.mines-paristech.fr/index.php/s/HAWmw7Fs927EtME

*Data availability.* DSSI results derived from the implementation of Heliosat-V for validation on all 11 stations are available at https://cloud.

mines-paristech.fr/index.php/s/HAWmw7Fs927EtME, along with simulated and MSG measured reflectances, cloud and clear-sky indices and clear-sky irradiance estimates from the McClear model. The manually filtered clear-sky instants are also provided for all 11 locations.

## Appendix A: Methods

## A1   Set-up of validation

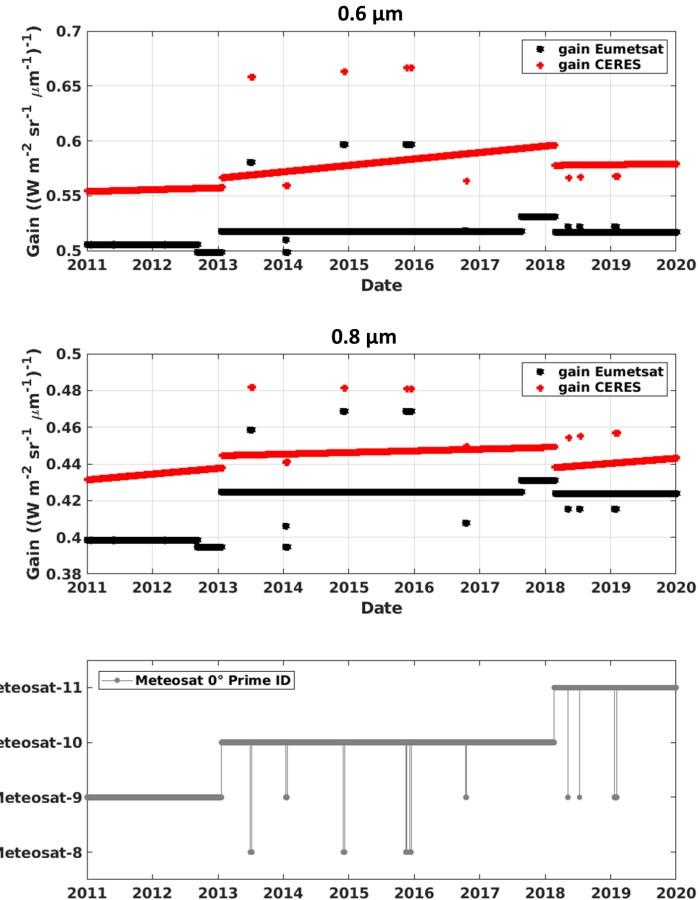

**Figure A1.** First two rows: calibration gains provided by EUMETSAT (black stars) and by CERES-SYN1deg (Doelling et al., 2018) (red stars) for 0.6 $\mu$m channel (first row) and 0.8 $\mu$m (second row) of SEVIRI sensor aboard Meteosat-9, between 2011 and 2019. Third row: ID of the operational satellite at the longitude 0°.

## Appendix B: Results

**B1   Simulated TOA clear-sky reflectances $\rho_{clear}$**

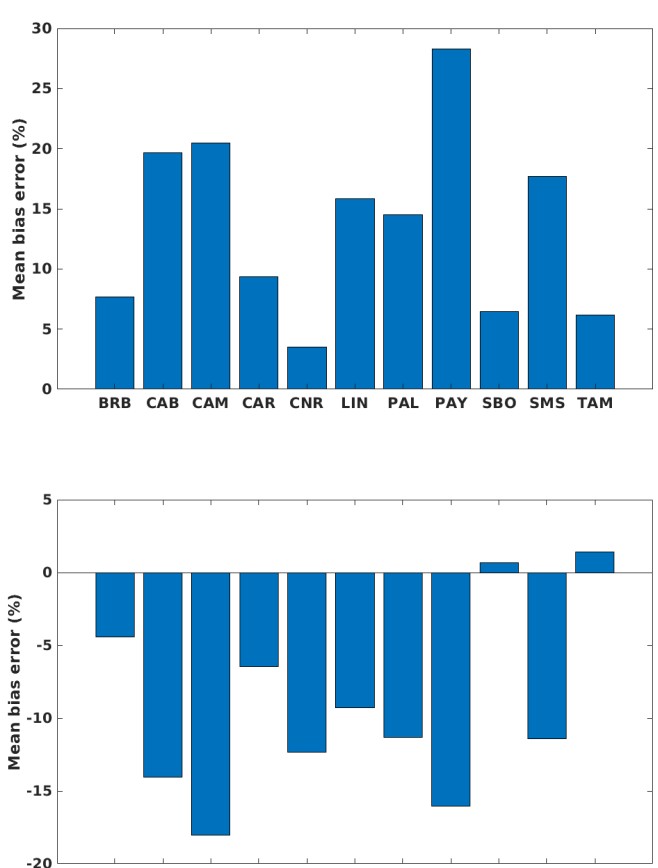

**Figure B1.** Relative mean bias errors of simulated clear-sky reflectances at the top of the atmosphere $\rho_{\text{clear}}$ ((simulation-measurement)/measurement) for channels 0.6 $\mu$m (upper panel) and 0.8 $\mu$m (lower panel), and for each BSRN site.

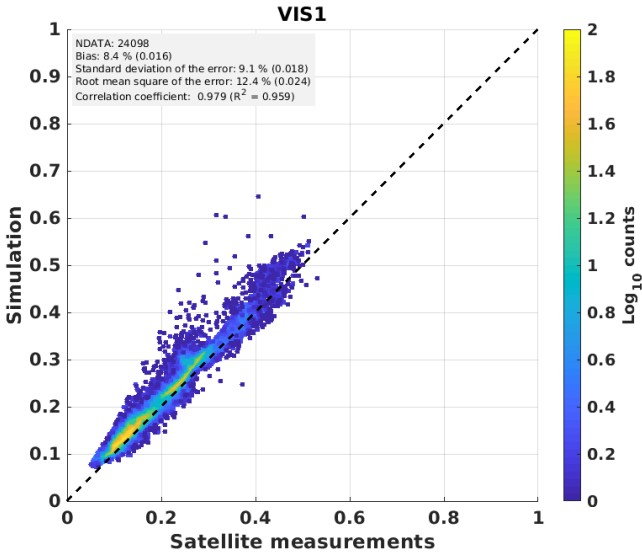

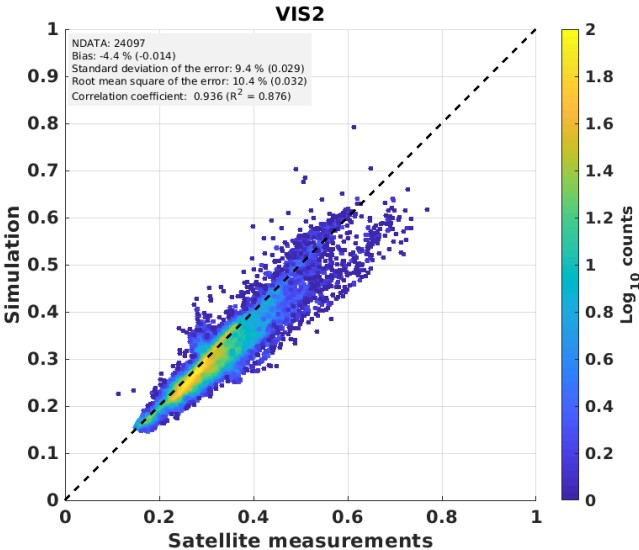

**Figure B2.** Simulation of clear-sky reflectances at the top of the atmosphere ($\rho_{\text{clear}}$) for MSG 0.6 $\mu$m (upper panel) and 0.8 $\mu$m (lower panel) spectral channels compared with actual satellite measurements. The comparison is done for all 11 locations, for the year 2011. Only instants with BRDF data of best quality are used (quality flag 0 of MCD43C1, "Best quality, 100% with full inversion")

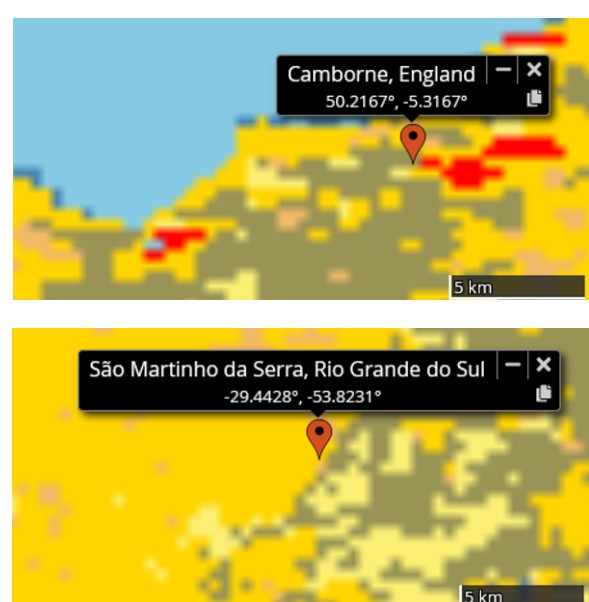

**Figure B3.** Land cover types around measurement stations São Martinho da Serra (Brazil, upper panel) and Camborne (United Kingdom, lower panel) for 2011. In red: urban and built-up lands; in gray: croplands/natural vegetation mosaics; in light yellow: croplands; in dark yellow: savanna; in beige: grasslands; in blue: water bodies. Data from Terra + Aqua MODIS product MCD12Q1 version 6, following the International Geosphere-Biosphere Programme classification scheme. Credit: NASA Worldview

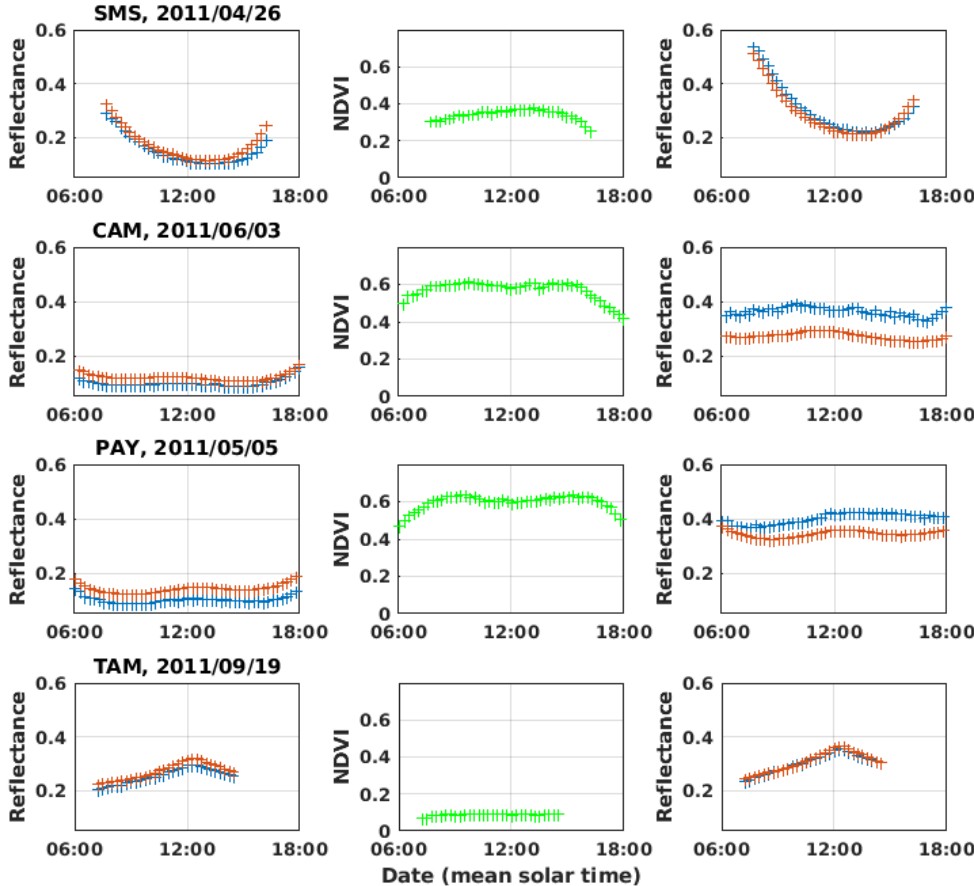

**Figure B4.** Comparison between simulated (red plus signs) and measured reflectances (blue plus signs) at the top of the atmosphere for one day in clear-sky conditions, for 0.6 $\mu$m (first column) and 0.8 $\mu$m channels (third column). NDVI computed from satellite measurements is shown on second column. Rows from top to bottom: locations of São Martinho Da Serra, Camborne, Payerne and Tamanrasset BSRN stations.

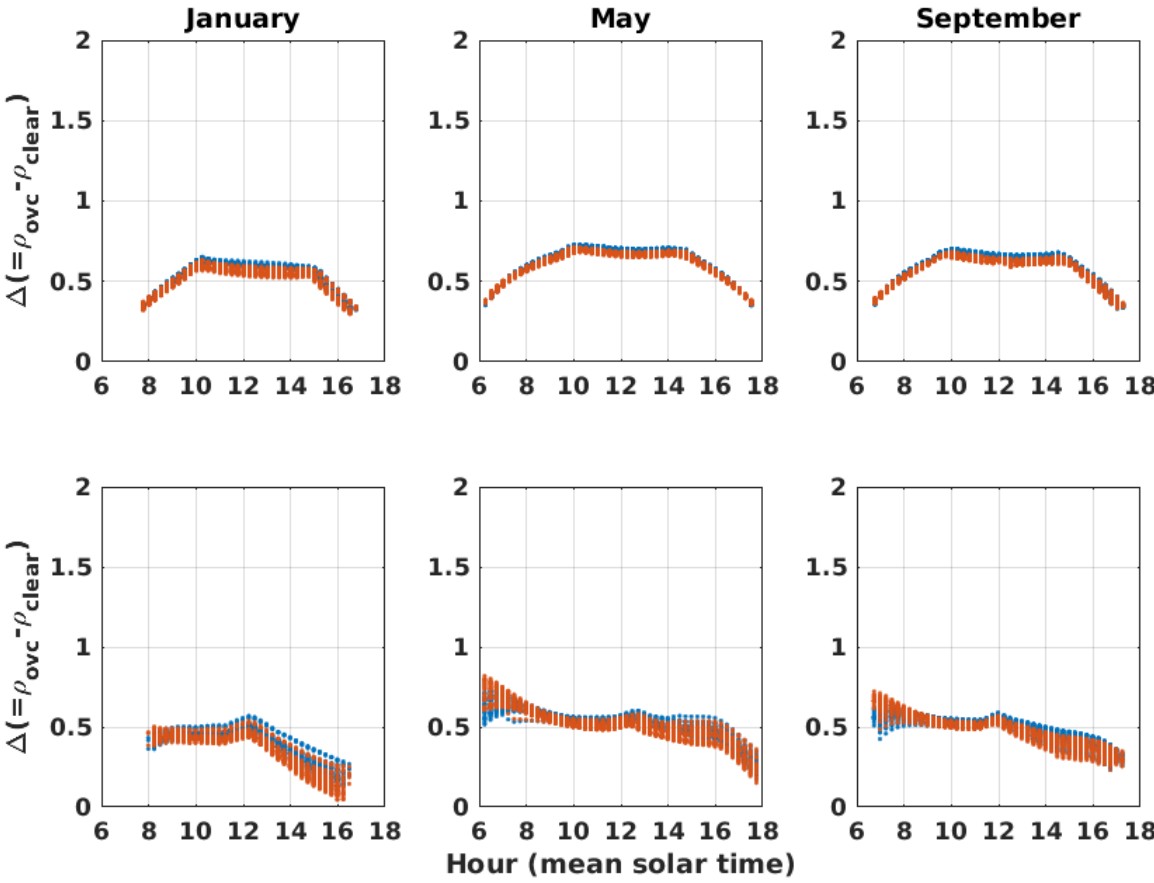

**Figure B5.** Difference between simulated reflectances at the top of the atmosphere in overcast and in clear-sky conditions $\Delta = \rho_{ovc} - \rho_{clear}$ for Tamanrasset (Algeria, upper row) and Sede Boker (Israel, lower row) locations and for January, May and September calendar months (three columns from left to right). In blue dots: MSG 0.6 $\mu$m channel ; in red dots: MSG 0.8 $\mu$m channel.

**Table B1.** Validation results for 15-min means of all-sky DSSI, for the year 2011. Results based on the imagery of Meteosat-9/SEVIRI 0.8 $\mu$m channel.

| Station | Number of samples | Mean BSRN W m$^{-2}$ | Bias W m$^{-2}$ (%) | RMSD W m$^{-2}$ (%) | Correlation coefficient (R) |
|---|---|---|---|---|---|
| Brasilia | 13570 | 504 | 13 (3) | 142 (28) | 0.871 |
| Cabauw | 13222 | 301 | -27 (-9) | 93 (31) | 0.919 |
| Camborne | 12731 | 310 | -28 (-9) | 117 (38) | 0.875 |
| Carpentras | 12642 | 452 | 19 (4) | 81 (18) | 0.958 |
| CENER | 14164 | 412 | -9 (-2) | 96 (23) | 0.932 |
| Lindenberg | 13637 | 317 | -18 (-6) | 92 (29) | 0.920 |
| Palaiseau | 13993 | 335 | -14 (4) | 88 (26) | 0.934 |
| Payerne | 9191 | 387 | -22 (-6) | 99 (26) | 0.936 |
| Sede Boker | 13574 | 589 | 23 (4) | 91 (16) | 0.947 |
| Sao Martinho da Serra | 5864 | 501 | -49 (-10) | 124 (25) | 0.918 |
| Tamanrasset | 13609 | 579 | 15 (3) | 89 (15) | 0.954 |
| Total | 136197 | 424 | -6 (-2) | 101 (24) | 0.934 |

## B2 Comparison of satellite-based estimates with ground-based measurements

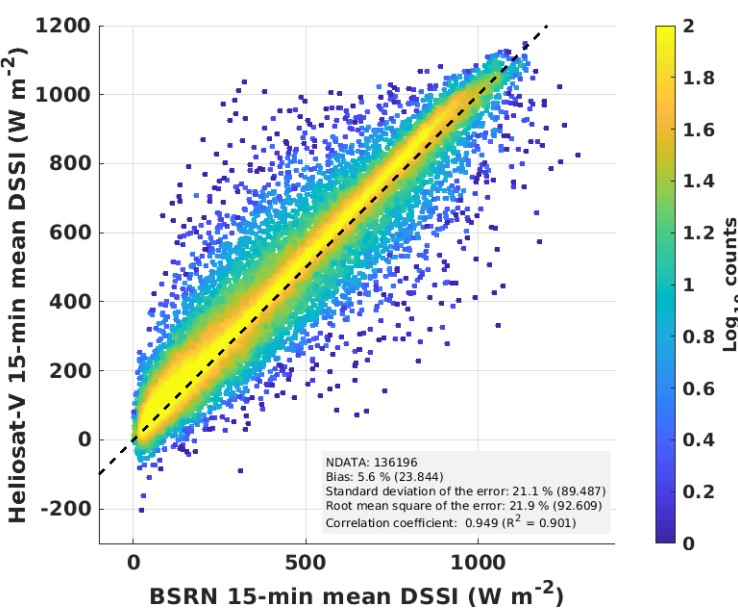

**Figure B6.** Impact of cloud phase on DSSI estimates. 2D-histogram of satellite-based DSSI estimates from the Heliosat-V method versus ground-based BSRN measurements for the MSG 0.6 $\mu$m channel. The liquid cloud look-up table of overcast-sky TOA reflectances is replaced for the ice cloud LUT.

*Author contributions.* Conceptualization by BT, YMSD and PB. Investigation, validation and writing -original draft by BT. Visualization by BT and PB. BT and BG did software, with contributions from PB. Supervision by PB and BG. Methodology by BT, PB and YMSD. All authors brought contributions in the writing process.

*Competing interests.* BT is funded by Region Sud and the company Transvalor for his PhD work.

*Acknowledgements.* We would like to thank operators of the BSRN stations for producing and providing precious validation measurements. We are grateful to the libRadtran team for their open radiative transfer model and interactions. The MODIS MCD43C1 version 6 data product was retrieved from the online NASA Earthdata Search, courtesy of the NASA EOSDIS Land Processes Distributed Active Archive Center (LP DAAC), USGS/Earth Resources Observation and Science (EROS) Center, Sioux Falls, South Dakota, https://search.earthdata.nasa.gov/.
We also acknowledge EUMETSAT and the ECMWF for providing respectively Meteosat-9 and MACC reanalysis aerosol, water vapour and ozone data. We acknowledge the use of imagery from the NASA Worldview application (https://worldview.earthdata.nasa.gov), part of the NASA Earth Observing System Data and Information System (EOSDIS). We finally acknowledge Olivier Boucher, Alba Lorente, François-Marie Bréon, Jérome Vidot, Nicolas Ferlay, Philippe Dubuisson, Maximilien Patou, Alexander Marshak, Alexei Lyapustin, Jay Herman, Yuekui Yang, Tamas Varnai, Richard Perez, Paul Stackhouse, Seiji Kato, David Doelling and Bastiaan Van Diedenhoven for useful
discussions.

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
