# Peer review of "An alternative cloud index for estimating downwelling surface solar irradiance from various satellite imagers in the framework of a Heliosat-V method"

_Atmospheric Measurement Techniques, 2020_

## Referee Comment (RC3)

**Review of Manuscript amt-2020-480**

An improved cloud index for estimating downwelling surface solar irradiance from various satellite imagers in the framework of a Heliosat-V method By Benoît Tournadre, Benoît Gschwind, Yves-Marie Saint-Drenan, and Philippe Blanc

**Background**

The study is in the framework of the development of the HeliosatV method for estimating downwelling solar irradiance at the surface from satellite imagery. It is claimed that a new way to retrieve a cloud index from a large variety of satellite instruments on geostationary and non-geostationary platforms was developed. The method uses simulations from a fast-radiative transfer model to estimate overcast (cloudy) and clear-sky (cloud-free) satellite scenes of the Earth's reflectance. An implementation of the method is applied to the visible imagery from a Meteosat Second Generation satellite. Results from preliminary implementation of Heliosat-V and ground-based measurements show a correlation coefficient reaching 0.948, for 15-minute means of downwelling surface radiation, similar to operational and corrected satellite-based data products (0.950 for HelioClim3 version 5 and 0.937 for CAMS Radiation Service).

**General Comments**

- 1. It was difficult to read the paper due to lack of transparency caused by following:
  - a) Superfluous information dominates the text.
  - b) This is not a review paper so there needs to be a strong focus on the objective of the paper.
  - c) Many statements were repeated several times in the text.
  - d) There was a frequent jump from one topic to another.
- 2. The discussion in many instances went into detail on a special topic (like aerosols) that were not utilized in implementing the methodology. One wonders why dwell on it.
- 3. If a new methodology is proposed there is a need to demonstrate that it is better than anything else that is available. The Authors state in the Abstract:

Results from our preliminary implementation of Heliosat-V and ground-based measurements show a correlation coefficient reaching 0.948, for 15-minute means of DSSI, similar to operational and corrected satellite-based data products (0.950 for HelioClim3 version 5 and 0.937 for CAMS Radiation Service). Since improvement was not demonstrated (against an earlier version of their methodology or any other methodology) why would one be interested in the described approach? Moreover, why do they provide information on the correlation only?

- 4. Something is amiss in the logic of the approach: the Heliosat idea is to use a cloud index to get Downwelling surface solar irradiance (DSSI). This, for simplicity of the process and contrary to the LUP table approach that is based on simulations. In order to use the LUP tables one needs to know the parameters used in the simulations to do the matching with the observed TOA radiance/albedo. Not clear what is the benefit in doing the simulations that are not appropriately utilized?
- 5. The argument that the simulated SAL is better than the library of min SALs or that it can be used with every satellite, is weak. To estimate the DSSI for each case using the Heliosat approach one needs the SAL at the time of the observation. How is such matching achieved?

**Some misleading and unsubstantiated statements:**

It is stated:

"the lower boundary is "archive-based", in most literature we reviewed: it is a minimum based on a time series of past satellite imagery. Such an approach is hardly applicable to non-geostationary satellites due to variable viewing geometries and a low revisit time

In this paper, we aim at finding an alternative to the need for archives of satellite imagery. It would then be easier to consider imagery from non-geostationary spaceborne platforms and produce a worldwide coverage.

It was not shown how the simulated albedo is used in the context of geostationary satellites and/or polar orbiters.

Stated:

Heliosat-V is a method approximating the attenuation of DSSI radiation by clouds with a cloud index, n.

We aim at developing an alternative "stateless" method to extend the application field of the cloud-index approach to a wider variety of orbits and optical shortwave sensors

What is "stateless"? How was it extended to polar orbiters? The paper deals only with SEVIRI.

Briefly, in addition to the lack of clarity of the text it seems that it was not demonstrated that the stated objectives of improvement and generalization have been achieved.

In the section between lines 245-250 the following statements are made:

 The use of optimal calibration is out of the scope of our work. Still, we compared gains coefficients proposed by EUMETSAT gEUM with those provided by Doelling et al. (2018) gD2018 for the measurements produced by the Meteosat-9 250 0.6 and 0.8 μm channels in 2011. They show a mean disagreement, calculated as (gEUM – gD2018)/gD2018, of about -9 % for 0.6 μm and -8 % for 0.8 μm during this period (also illustrated on Fig. A1). Such errors will affect with the same magnitude the agreement between numerical simulations and measurements of clear-sky TOA reflectances, underlining the importance of absolute calibration for the Heliosat-V method.

Not obvious what is the message of the Authors here: on one hand, the calibration is out of the scope of their work. Then they report on the evaluation of different gains which show large differences (-9 %). They continue to state:

Such errors will affect with the same magnitude the agreement between numerical simulations and measurements of clear-sky TOA reflectance, underlining the importance of absolute calibration for the Heliosat-V method.

Which is it? Is it important or not?

**In Figure 6 provided are:**

Simulation of clear-sky reflectances at the TOA ( $\rho$ clear) for MSG 0.6  $\mu$ m (left panel) and 0.8  $\mu$ m (right panel) spectral channels compared with actual satellite measurements. The comparison is done for all 11 locations, for the year 2011.

How was this comparison done? At each of the 11 locations, the atmospheric conditions are different. The atmospheric correction would be different. Not clear how the comparison was performed.

In summary, this manuscript is not ready for publication.

---

## Author Comment (AC1)

Response to comments from Reviewer 1:

We thank the reviewer for the useful comments, analyses and proposed corrections. We humbly apologize for the inconvenience caused by repeated delays in our response since the publication of reviews.

Tournadre et al. developed a new way to estimate downwelling surface solar irradiances (DSSI) from satellite images for Heliosat-V. Similar to previous Heliosat algorithms, the cloud index is needed in the DSSI estimation. In this new method, the maximum and minimum reflectances needed in the cloud index calculations are simulated using radiative transfer model instead of taking from archives of satellite images. The authors have demonstrated that the DSSI derived using this new method have good agreement with the CAMS and HelioClim3 DSSI. The new method is very promising. It has the advantage to be applied to both geostationary and polar orbiting satellites to get a global consistent DSSI data set using the same algorithm. The long term global DSSI data set will be interested by the solar energy and climate related communities. The authors have described the algorithm and results clearly. I think it is a good paper for AMT.

Specific comments

1) Line 19, ' plus a diffuse component due to scattering caused by the atmosphere (clouds, gases, aerosols) … '

Please also add 'absorption' in the sentence. In Fig. 2 you showed the gas absorptions by O2, O3, H2O.

> The purpose of the sentence is to specify that we only look at the hemispherical integral of the radiation reaching the surface and not its decomposition in beam and diffuse components. We propose to clarify by "*DSSI considers the irradiance coming from all directions of the hemisphere above the surface: the irradiance coming from the direction of the Sun, usually referred to as beam horizontal irradiance, plus a diffuse component due to scattering caused by the atmosphere (clouds, gases, aerosols) and reflection by the surface, usually referred to as diffuse horizontal irradiance.*"

2) Line 22 'renewable solar energy industries, …'

Is 'renewable' needed here?

> We removed the term.

3) Line 41-42. This sentence can be combined with the paragraph from Line 43.

> We applied the modification.

4) Line 50, Please add the following paper in the reference list because they also use cloud properties to derive DSSI.

Retrieval and validation of global, direct, and diffuse irradiance derived from SEVIRI satellite observations

Greuell J. F. Meirink P. Wang  https://doi.org/10.1002/jgrd.50194

> We added the reference Greuell et al. (2013).

5) Line 100-101 , 'the upper boundary variables Xmax and Xmin'

Change to 'the upper and lower boundaries …. '

> We added the missing elements.

6) Line 165 . This paragraph describes the MACC reanalysis used in the LUT. It is not clear if the MACC reanalysis has day, monthly or yearly AOD and which AOD is used.

> The following sentences are added: "*Data from MACC are extracted from the McClear service (http://www.soda-pro.com/web-services/radiation/cams-mcclear). MACC values are originally given on a 3-hour time step and with a spatial resolution of about 80 km (Inness et al. 2013; Lefèvre et al., 2013). The McClear service applies to MACC data a bilinear spatial interpolation onto the considered location, and a linear interpolation in time to a 1-min time step (Lefevre et al., 2013).*"

7) Lines 184-185 can be combined with the paragraph below it.

> We merged two paragraphs.

8) Line 212 ' Heliomont' Is it a typo?

> We added the uppercase correction "HelioMont" (which is the algorithm described in Stöckli (2014))

9) Line 233 "ant", typo?

> We corrected the typo.

10) Table1. What are the cloud base heights?

> We added cloud base heights in Table 1: 200 m for low thick cloud, 2 km for high thick cloud.

Please also add a table for the clear-sky LUT, including the BRDF, aerosols settings etc.. It is not complete if only having the table for the cloud LUT.

> We added such a table. Infos sur la resolution de srtm nécessaire → benoit g.

11) Line 258 '….for solar zenith angles lower than 80°'

Why do you use solar zenith angle until 80 degree in the validation? In the LUT, the solar zenith angle is until 85 degree. Is it possible to extend the solar zenith angle until 90 degree in the LUT?

We change this sentence for: "*As the work is exploratory on a new method, we limit ourselves to conservative situations with solar zenith angle lower than 80°, covering most cases. For higher SZA, some effects not considered by the method can occur, including shadowing and high parallax effects.*"

It would be possible to extend the LUT until SZA=90°. Since the preprint submission, we extended the LUT until 88°. For this paper, we considered 85° sufficient as it is already beyond the range used for validation.

12) Line 268 'However reflectance in the near infrared 0.8 μm channel are significantly higher, so is the absolute value of STD.' Readers might want some explanations why the reflectances at 0.8 micron channel is larger than the 0.6 micron channel. Actually it is explained in the discussion section. This happens also in other paragraphs in the results section.

The sentence line 268 has been removed because oversimplify the description. Part 3 has been reorganized to emphasize the different origins of errors on the computation of the cloud index, notably with the following paragraph:

*"The validity of cloud index components, $\rho_{sat}$, $\rho_{clear}$, and $\rho_{ovc}$, defines the accuracy of n. From Equation (3), the uncertainty on the cloud index can be written as:*

$$\delta n = \left( \frac{\partial n}{\partial \rho_{\text{sat}}} \right) \delta \rho_{\text{sat}} + \left( \frac{\partial n}{\partial \rho_{\text{clear}}} \right) \delta \rho_{\text{clear}} + \left( \frac{\partial n}{\partial \rho_{\text{ovc}}} \right) \delta \rho_{\text{ovc}} \tag{9}$$

This leads to

$$\delta n = \frac{1}{\Delta} \left( \delta \rho_{\text{sat}} - (1-n)\, \delta \rho_{\text{clear}} - n\, \delta \rho_{\text{ovc}} \right) \tag{10}$$

*Where $\Delta = \rho_{ovc} - \rho_{clear}$. It appears that the "clear-sky error" $(1-n)\, \delta\rho_{clear}$ will be more significant in clear-sky conditions (i.e., n is close to 0), and the "overcast-sky error" $n\, \delta\rho_{ovc}$ will be more important in overcast conditions (i.e., n is close to 1). Besides, the error on cloud index will be inversely proportional to $\Delta$, the difference between overcast and clear-sky TOA reflectances. Because of this relationship between the errors on cloud index and reflectances, the discussions in this section are focused on absolute values of reflectance errors."*

The origin and impact of the contrast between overcast and clear-sky reflectances are illustrated in additional figures and discussed in section 3.1.4:

*"The difference $\Delta$ between overcast and clear-sky reflectances is bigger when the overcast reflectance is relatively low and clear-sky reflectance is relatively high. High values of $\Delta$ mean a good quality of cloud index estimation (cf. Equation 10). We study the dependencies of $\Delta$ with the simulated reflectances to identify conditions that will cause high uncertainties on the computation of the cloud index. In general, we observe that the computed value of $\Delta$ is higher for the 0.6 μm channel than for 0.8 μm, as a combination of surface, cloud and clear atmosphere spectral signatures. This is illustrated in Figure 9 for stations SMS and CAM. We observe however for the desert stations TAM and SBO that both channels present similar values of $\Delta$ (Fig. B4). $\Delta$ depends also on the viewing and solar geometries because of $\rho_{ovc}$ and $\rho_{clear}$ different angular signatures. It leads for example for SMS station and channel 0.8 μm to low values of $\Delta$ in January morning and high values of $\Delta$ in the evening, which can be*

*explained by the strong forward scattering of clouds occuring in these conditions."*

[Figure]

**Figure B3.** Land cover types around measurement stations São Martinho da Serra (Brazil, upper panel) and Camborne (United Kingdom, lower panel) for 2011. In red: urban and built-up lands; in gray: croplands/natural vegetation mosaics; in light yellow: croplands; in dark yellow: savanna; in beige: grasslands; in blue: water bodies. Data from Terra + Aqua MODIS product MCD12Q1 version 6, following the International Geosphere-Biosphere Programme classification scheme. Credit: NASA Worldview

[Figure]

**Figure 9.** Difference between simulated reflectances at the top of the atmosphere in overcast and in clear-sky conditions $\Delta = \rho_{\text{ovc}} - \rho_{\text{clear}}$ for São Martinho da Serra (Brazil, upper row) and Camborne (United Kingdom, lower row) locations and for January, May and September calendar months (three columns from left to right). In blue dots: MSG 0.6 $\mu$m channel ; in red dots: MSG 0.8 $\mu$m channel.

[Figure]

**Figure B4.** Difference between simulated reflectances at the top of the atmosphere in overcast and in clear-sky conditions $\Delta = \rho_{ovc} - \rho_{clear}$ for Tamanrasset (Algeria, upper row) and Sede Boker (Israel, lower row) locations and for January, May and September calendar months (three columns from left to right). In blue dots: MSG 0.6 $\mu$m channel ; in red dots: MSG 0.8 $\mu$m channel.

13) Lines 272 – 275. Figure 7 shows the results compared to measurements at the PAY and CAM SMS stations. Please provide some information about the surface type of the stations used in the figure. When it is clear-sky, the surface type, aerosols are more import.

A description of land cover type is added and a figure for SMS and CAM land cover types is provided in appendix (also below):

"*Both SMS and CAM are surrounded mainly by various types of vegetation and some urban area for the case of CAM (Figure B3).*"

We also add a discussion on the role of aerosols on reflectance variability simulated in clear-sky conditions in section 3.1.2.:

"*For CAM, some higher values of $\rho_{clear}$ are observed in January. This can be attributed to high aerosol optical depth during this period, as illustrated in Figure 8. It shows that $\rho_{clear}$ is not only sensitive to time variations of surface properties but also to atmospheric composition*

*changes*."

[Figure]

**Figure 8.** Blue plus signs: simulated reflectances at the top of the atmosphere in clear-sky conditions $\rho_{\text{clear}}$ in January 2011 at Camborne station (CAM) and for MSG 0.6 $\mu$m. Red line: aerosol optical depth at 635 nm used for simulations.

Please also note we focus the analysis of reflectance variabilities on SMS and CAM stations on Figure 7. PAY is mentioned in the text being the station with the highest mean bias for the channel 0.6 µm, but it is not represented on Figure 7.

14) In line 272, Figure 7 should be Figure 6.

Since BRDF is an important feature in the clear-sky LUT, it would be nice to show a figure at PAY, CAM, SMS with diurnal cycle for a clear-sky day. Please use 0.6 and 0.8 channel both when there are green grass on the ground surface.

The Figure 7 in the first submitted manuscript emphasizes the diurnal variations of reflectances, which are not an obvious signal in the 2D histograms of Figure 6. We add a figure in appendix showing the diurnal cycle for simulated and measured clear-sky reflectances (see below), referred to in the section:

"*We observe that simulations are able to reproduce partly the diurnal variability observed in clear-sky conditions (also refer to Figure B2 for channel 0.6 µm and 0.8 µm under different surface conditions)*"

[Figure]

**Figure B2.** Comparison between simulated (red plus signs) and measured reflectances (blue plus signs) at the top of the atmosphere for one day in clear-sky conditions, for 0.6 μm (first column) and 0.8 μm channels (third column). NDVI computed from satellite measurements is shown on second column. Rows from top to bottom: locations of São Martinho Da Serra, Camborne, Payerne and Tamanrasset BSRN stations.

15) Line 280 'Figure 6' should be Figure 7.

We corrected the error.

16) Fig. 7 Why the simulated reflectances have better agreements with measured reflectances at SMS than at CAM?

It might not be due to the calibration of MSG because it would have the same bias in the full disk image. It seems the ice cloud LUT has similar diurnal cycle to the 99 percentiles of the measurements but the simulated reflectances are larger than the measurements at CAM. It could be at CAM the cloud are less brighter than at SMS. Does it suggest the simulated maximum reflectance should depend on location?

We change the following sentence as the analyse is not sufficient to assess an agreement, "The first row of Figure 7 shows a good agreement between most reflective satellite scenes of the São Martinho da Serra pixel and $\rho_{ovc}$" and replace it saying, "Some patterns are similar in simulated $\rho_{ovc}$ and 99th percentile of measurement: in the forward scattering conditions (evening on the West edge of Meteosat disc) where both agree on increased values of $\rho_{ovc}$". It is difficult to assess biases of the simulated overcast reflectances because of significant uncertainties on calibration gains.

... This percentile approach is however not reliable in all cases. For example, at Sede Boker in May and September, we cannot consider that the 99[th] percentile correspond to thick clouds conditions, because of the very dry climate and low reflectances observed.

17) Line 320.

Fig. C1. Why there are some outliers with large reflectances in McClear? Is it due to the model or the aerosol data? I would expect the outliers on two sides of the 1:1 line.

McClear is the model computing surface solar irradiance in clear-sky conditions notably from the description of atmospheric composition in aerosols, ozone, water vapor etc. (see section "The clear-sky model of surface irradiance McClear"). The outliers are probably due to some cloud contamination that were not identified in the satellite measurements. We prefer to remove the figure, as these outliers could influence the mentioned "3%" value of McClear's bias. We refer to McClear v3 publication Gschwind et al. (2019) to discuss its uncertainties: "Gschwind et al. (2019) report for example relative mean biases of the McClear model from -3.6% (Barrow, Alaska, USA) to +3.2% (Payerne, Switzerland), when compared to BSRN irradiance measurements."

Please note also that the Discussion section is merged with Results section in order to present more smoothly.

18) The authors did not mention direct irradiances in the paper. Are there any plans about the DNI?

The estimation of direct irradiance would sure be of interest. For now, we are focusing on estimating the cloud index.

---

## Author Comment (AC2)

Response to comments from Reviewer 2:

We thank the reviewer for the useful comments, analyses and proposed corrections. We humbly apologize for the inconvenience caused by repeated delays in our response since the publication of reviews.

General comments:

The explicit strength of the "original" Heliosat approach (referred to as Heliosat-o in this review) is that the retrieved cloud index ("cloud transmission") is completely based on observations. No simulations or external data are needed to retrieve the cloud index (cloud transmission) but the observed radiances are used. This includes the retrieval of Xmin ("clear sky reflection") & Xmax ("calibration"). Heliosat-o and the resulting radiation data are well validated and established (e.g CM SAF, ISE, University of Oldenburg and Bergen, Satellight….) and already close to the accuracy of well maintained ground based stations. Of course, there are some limitations linked with the Xmin retrieval, as listed by the authors (L85). However, some of the mentioned handicaps are already partly resolved (e.g. shadow correction method by University of Oldenburg) or on average of relative small effect (e.g. long lasting clouds occur usually in the North-West during wintertime. This means high COD and low SZA. Hence, low solar irradiance and thus low absolute errors induced by uncertainties in Xmin). In my opinion there is a high likelihood that the simulation of Xmin adds more handicaps and uncertainties than it resolves. Thus, the central question is: Is there an overall benefit, concerning accuracy and precision, if the observational-based Xmin retrieval is replaced by simulations. Why should the simulations lead on average to more accurate results than using observations ? The authors mention "Simulations consider the anisotropy of the reflectances caused by both surface and atmosphere, and are adapted to the spectral sensitivity of the sensor. The anisotropy of ground reflectances is described by a bidirectional reflectance distribution function model and external satellite-derived data". Simulations might consider it, but to my experience they induce also additional uncertainties, e.g. the uncertainty induced by using $3^{rd}$ party surface albedo data can easily lead to a bias of several per cent. Further, as for RMIN, clear sky situations are needed to retrieve the surface albedo, thus concerning long lasting clouds the same handicap is shared. The needed BRDF (ADM) functions induce further uncertainties and add complexity. A more complex method providing overall a lower accuracy would be of no significant value. The effect of SAL (surface albedo) and BRDF is already considered by observational-based Xmin for the same sensor and viewing geometry, no need for simulation.

We appreciate your complementary views on Heliosat methods. We agree on the strength (especially the simplicity) of a self-calibrated method like Heliosat-o to produce climate data records like SARAH over several generations of satellite sensors, especially for past sensors like Meteosat First Generation/MVIRI.

To answer the central question "Is there an overall benefit, concerning accuracy and precision, if the observational-based Xmin retrieval is replaced by simulations?", the method already shows a precision similar to operational products. But developing this new method aims also at investigating on the origin of cloud index uncertainties. Using simulations of TOA reflectances integrating surface, clear atmosphere, and cloud properties provides flexibility for future improvements and sensitivity analyses. We also consider that the development of an alternative method to compute the cloud index with different assumptions is useful to assess, for example, the robustness of DSSI time variations within multi-model comparison exercises.

This work is exploratory, and our publication comes as a first version showing encouraging results. Several significant sources of errors are identified (source of calibration gains, spectral interpolation of MODIS BRDF data, cloud properties used in the $\rho_{ovc}$ Look-Up Table, angular description of the LUT). These errors will be further considered in future works (including on-going works), and their treatment is likely to improve results. Please also note that, as the paper focuses on the computation of the cloud index, the clear-sky index/cloud-index relationship is not investigated, and may also improve the quality of future results.

Concerning surface BRDF, products like MODIS MCD43C1 are useful to consider surface anisotropic reflection and reproduce TOA reflectances in clear-sky conditions. It provides a spectral description that allow us to apply the same method to a wide range of satellite sensors and channels, and a global coverage so we can apply the method to satellites on different orbits with the same inputs. We agree that using external sources of data to describe the surface BRDF introduces sources of errors, linked to data product quality (here MODIS MCD43C1), the BRDF model (RossThick-LiSparse) and their integration into our radiative transfer simulations. Despite all these sources of errors, we observe that our simulations are sufficiently close to clear-sky observations to ensure a good quality of surface irradiance estimates.

Concerning the long-lasting clouds situations, MODIS MCD43 is based on TOA reflectance measurements that passed a detailed cloud masking procedure involving about 22 spectral channels from visible to thermal infrared (Ackerman et al., 2010) and an atmospheric correction. In case of long-lasting clouds, the product relies on back-up information to provide information on BRDF. This is information may be of lower quality, but without cloud contamination.

Concerning the use of radiative transfer simulations to compute the cloud index and as previously discussed, modeling clear-sky and overcast reflectances with knowledge on surface, clear-sky atmosphere and clouds is a way to identify and quantify sources of errors in cloud-index methods. An example of potential interest is for overcast conditions: Heliosat-2 considers it as only depending on solar zenith angle (Rigollier et al., 2004). For heliosat-o as described in Mueller et al. 2012 & 2015, the $X_{max}$ value corresponds to a percentile on an archive of measurements for a given region and does not appear to depend on the sun-satellite geometry. This is mentioned in SARAH-2's ATBD as a potential source of error. A method like ours could be used to assess the uncertainty caused by the assumption of a Lambertian cloud. Another example is that methods using the cloud-index approach have difficulties to isolate the irradiance attenuation due to aerosols: aerosols loads generally increase the top-of-atmosphere reflectance and thus can be erroneously detected as clouds in the cloud-index computation. In the meantime, they lead to decrease the modeled clear-sky surface irradiance, having aerosol information in input (Mueller et al. (2015)). In Heliosat-V scheme, the variability of aerosols is used both in simulation of clear-sky TOA reflectance and in the clear-sky surface irradiance model e.g. McClear. It should limit this source of error, even if we still have to explore it.

Major concerns:

In my opinion the authors fail to show the advantage of combining the Heliosat relation (equation) with simulations of the radiances in order to get Xmin ("clear sky

reflection"). If radiances (reflectances) are simulated than why not simply using one of the several RTM based LUT approaches or ECMWF.  By the way, using BRDFs simulations to estimate radiances observed by satellite is already applied since decades in RTM based LUT approaches,  thus this Is not a new idea. Where is the benefit to use the Heliosat relation (equation 1) when the special  strength of Heliosat is disminished by using simulations ?  These questions are not appropriately addressed in the manuscript. The authors mention that a motivation for the approach is the use of polar orbiting satellites, but again what is the advantage compared to RTM based LUT approaches (using COD&reff or TOA Albedo).

The interest for a cloud-index method using radiative transfer simulations is addressed above, and we will add more discussion to address this point. To summarize, we consider that this method is also of interest to explore possibilities, being different from previous cloud-index methods and full RTM based LUT approaches, and flexible for sensitivity analyses. Heliosat-o is very useful as a full measurement approach, but we consider also of interest exploring a new cloud-index method able to improve e.g., the description of aerosols and of cloud anisotropy, and even to be applied to non-geostationary sensors.

The introduction has been reorganized and completed to address these concerns.

In summary, a more thorough discussion and description of the pros and cons of the presented method compared to established methods should be added (Heliosat-o and RTM LUT approaches). Uncertainties of BRDF and SAL should be discussed, more information on SAL source should be added.

We add a figure in appendix showing improved results comparing clear-sky measurements of reflectance and simulations made using only the BRDF with the best quality (figure reproduced below)

[Figure]

[Figure]

**Figure B2.** Simulation of clear-sky reflectances at the top of the atmosphere ($\rho_{clear}$) for MSG 0.6 $\mu$m (left panel) and 0.8 $\mu$m (right panel) spectral channels compared with actual satellite measurements. The comparison is done for all 11 locations, for the year 2011. Only instants with BRDF data of best quality are used (quality flag 0 of MCD43C1, "Best quality, 100% with full inversion")

Also the solar zenith angle dependency of SAL in relation to BRDF should be discussed in more detail.

We add the following description in the Results section: "On Figure 8, we compare $\rho_{clear}$ values with the surface reflectance $\rho_{surface}$, computed with the RossThick-LiSparse model applied to BRDF parameters derived from MODIS 646 nm channel, and using viewing and solar geometries considered. Firstly, we see that $\rho_{clear}$ values are significantly higher than $\rho_{surface}$ with a different diurnal pattern. This shows the importance of considering the atmosphere anisotropic reflectance to reproduce TOA reflectances. We also can see the contribution from the surface anisotropy in the $\rho_{clear}$ simulations. This appears in particular close to the backscattering direction where surface reflectance is enhanced: around noon in Camborne and the morning in São Martinho da Serra." This comes with the following figure:

[Figure]

**Figure 8.** Comparison between simulations of clear-sky reflectances at the top of the atmosphere for MSG 0.6 $\mu$m channel ($\rho_{clear}$, blue plus signs) and corresponding surface reflectances computed with the RossThick-LiSparse model applied to MODIS MCD43C1v6 BRDF parameters for the channel 646 nm ($\rho_{surface}$, red plus signs) for five days in June 2011. Left panel: Camborne station (CAM) ; right panel: São Martinho da Serra station (SMS).

Further, the potential improvements should be proven and discussed thoroughly by comparison with established high quality data sets, which are using the original observational-based Heliosat-o approach and with other data sets from external sources, e.g. ECMWF. Please note, comparison with Helioclim might be not a real benchmark for improvements, see e.g. Posselt et al, Remote Sensing of Environment Vol 118, 2012, pp, 186–198.

> We use data from HelioClim3 and CAMS-RAD for comparisons. Posselt 2012 consider HelioClim1 which is no longer produced. The quality of HelioClim3 is similar to other operational products (e.g. Ineichen 2015 for intercomparisons of satellite-based DSSI products). CAMS-RAD is an open database of DSSI operationally provided in CAMS and all validation data used in this article will be provided as supplementary information to the article.

 Respective open data sets are available for inter-comparison. Concerning polar orbiting satellites, results should be compared to the ECWMF radiation data set.

> The investigation on non-geostationary sensors is on-going and will be treated in a future paper. ERA5 hourly values of DSSI are part of the data used for this extended validation.

I think that simulations of Rmin has been already used for the so called "Heliosat-2" version. Thus, the novel aspects of the approach should be reflected in more detail relative to "Heliosat-2" as well.

> Heliosat-2 approach is almost fully measurement based. It applies an atmospheric correction to estimate $X_{min}$ from an archive of imagery. For Xmax, it is an empirical model based on Nimbus-7 ERB observations. The Heliosat-V approach is therefore not so similar to Heliosat-2.

By the way, calling a method with Rmin simulation still Heliosat is quite confusing. Rmin simulation breaks with the basic idea of Heliosat, thus using the name Helioat should be avoided in order to avoid misleading interpretations.

> The Heliosat name only refers to the use of satellite data to estimate solar irradiance and to its link to Mines ParisTech, which has been involved in all Heliosat projects since 1980s. The Heliosat-4 method, used to produce the CAMS Radiation Service and McClear products, is based on radiative transfer simulations, information about aerosol, water vapor and ozone from CAMS and satellite-based products of cloud physical properties.

Overall the discussion should be modified to be more balanced and reflected , lessons learnt in other projects and communities  should be considered.

> We modify and complete the discussions. To facilitate the reading, the content of the discussion section is integrated to the "Results" different subsections dedicated to simulated and measured reflectances and to comparison of GHI estimates with external datasets.

Specific comments.

- Please change the title, improved is not prooven, see general comments.

  We modify the title for "An alternative cloud index for estimating downwelling surface solar irradiance from various satellite imagers in the framework of a Heliosat-V method"

- 70 „raw satellite numerical counts (Pfeifroth et al., 2017; Perez et al., 2002)";

Here and throughout oft he manuscript. Misleading citations. Raw satellite counts has been used already decades before within the Heliosat community. Please modify accordingly. In general ATBD, PUMs are grey literature. Please check the citations and replace them with peer reviewed articles where possible.

  We change the reference Pfeifroth et al. 2017 for Müller et al. 2015, and add the reference Cano et al. 1986.

- 80 "In this paper, we aim at finding an alternative to the need for archives of satellite imagery."

  We complete this sentence with previously mentioned arguments:

This is misleading, as long as radiances are needed using actual and/or 30 day is not a serious problem and not worth mentioning.

- 140 "Kc = 1−n introduced by Darnell et al. (1988)"

I think it is a well known and established that a modification for higher n is needed and respective modifications are published, please refer them.

  The paragraph has been slightly modified to consider the comment. We do not use one of those modification because they are generally based on observed non-linearity between clear-sky index and cloud index that may be partly caused to the treatment of $X_{max}$ in these methods. The new method partly resolved some of them, notably by considering the anisotropy of overcast reflectances with LUT.

- 190 "Cloud-index methods in the literature use various ways to estimate the TOA reflectances in overcast conditions (Perez et al., 2002; Lefèvre et al., 2007; Pfeifroth et al., 2017)."

Pfeifroth et al. 2017, again misleading citation. Please refer to the original peer-reviewed publications . In general ATBD, PUMs are grey literature.  Please check the citations and replace them with peer reviewed articles where possible.

We replace Pfeifroth et al. (2017) by Mueller et al. (2015)

- 65 Xmin is ued later on rho_clear please unify.

In equation (1), we use X for the variables, because depending on the study, the cloud index can be computed from TOA reflectances, albedos, raw numerical counts, bottom-of-atmosphere reflectances/albedos. We keep the rho notation for reflectances. We add around line 65 "We name these variables X as they can be of slightly different nature from one work to another (reflectance, albedo, radiance, digital count, etc.)."

**Citation**: https://doi.org/10.5194/amt-2020-480-RC2

Ackerman et al. 2010, DISCRIMINATING CLEAR-SKY FROM CLOUD WITH MODIS ALGORITHM THEORETICAL BASIS DOCUMENT (MOD35)

Ineichen, Pierre & Office fédéral de l'énergie OFEN, 2015 : Solar Resource Assessment and Forecasting, IEA SHC Task 46

---

## Author Comment (AC3)

Response to comments from Reviewer 3:

We thank the reviewer for the useful comments and remarks, and hope to address main concerns. We humbly apologize for the inconvenience caused by repeated delays in our response since the publication of reviews.

Background

The study is in the framework of the development of the HeliosatV method for estimating downwelling solar irradiance at the surface from satellite imagery. It is claimed that a new way to retrieve a cloud index from a large variety of satellite instruments on geostationary and non-geostationary platforms was developed. The method uses simulations from a fast-radiative transfer model to estimate overcast (cloudy) and clear-sky (cloud-free) satellite scenes of the Earth's reflectance. An implementation of the method is applied to the visible imagery from a Meteosat Second Generation satellite. Results from preliminary implementation of Heliosat-V and ground-based measurements show a correlation coefficient reaching 0.948, for 15-minute means of downwelling surface radiation, similar to operational

and corrected satellite-based data products (0.950 for HelioClim3 version 5 and 0.937 for CAMS Radiation Service).

General Comments

1. It was difficult to read the paper due to lack of transparency caused by following:

    a) Superfluous information dominates the text.

    b) This is not a review paper so there needs to be a strong focus on the objective of the

paper.

    We modified in depth the structure of the paper to make it easier to read, in particular the introduction and the results section, and removing information of secondary importance. We also rewrote the objective of the paper in the introduction as :

    "In this paper, we propose a cloud-index method based on radiative transfer modeling as an alternative to the archive-based approach. This exploratory direction aims at reproducing the satellite measurements of reflectances in both clear-sky and overcast conditions based on description of surface, clear atmosphere and cloud properties. Radiative transfer simulations are able to reproduce how TOA reflectances depend on viewing and solar geometries, with also their spectral distribution. In addition, it is possible to provide to the radiative transfer model input data that describes variations in space and time of clear atmosphere composition and of surface properties. Thus, our approach is useful to identify and quantify sources of errors in cloud-index methods. With a spectral and angular description, our method is also able to extend the application field of the cloud-index approach to a wider variety of orbits and optical shortwave sensors. In order to limit the effects of molecular scattering, ozone absorption and polarization present in the ultraviolet, and of the absorption of radiation by clouds in the near infrared, the method focuses on satellite imagery in the spectral range 400-1000 nm ($\lambda <$ 1000 nm). This range is wide enough to consider imagers on many meteorological satellites launched since the beginnings of spaceborne Earth observation."

c) Many statements were repeated several times in the text.

d) There was a frequent jump from one topic to another.

Thanks for noting these, the text has been modified keeping these remarks in mind.

2. The discussion in many instances went into detail on a special topic (like aerosols) that were not utilized in implementing the methodology. One wonders why dwell on it.

Aerosols are an important topic for the Heliosat-V cloud-index: we use aerosol data as input to simulate the clear-sky reflectances at the top of the atmosphere. We add a short discussion and the figure below on the effect of aerosols on TOA clear-sky reflectances in the Results section:

"For CAM, some higher values of $\rho_{clear}$ are observed in January. This can be attributed to high aerosol optical depth during this period, as illustrated in Figure 9. It shows that $\rho_{clear}$ is not only sensitive to time variations of surface properties but also to atmospheric composition changes."

[Figure]

**Figure 8.** Blue plus signs: simulated reflectances at the top of the atmosphere in clear-sky conditions $\rho_{clear}$ in January 2011 at Camborne station (CAM) and for MSG 0.6 $\mu$m. Red line: aerosol optical depth at 635 nm used for simulations.

3. If a new methodology is proposed there is a need to demonstrate that it is better than anything else that is available. The Authors state in the Abstract: *Results from our preliminary implementation of Heliosat-V and ground-based measurements show a correlation coefficient reaching 0.948, for 15-minute means of DSSI, similar to operational and corrected satellite-based data products (0.950 for HelioClim3 version 5 and 0.937 for CAMS Radiation Service).* Since improvement was not demonstrated (against an earlier version of their methodology or any other methodology) why would one be interested in the described approach?

In the current version of our method, we did not show improvement in terms of accuracy and precision in the validation results compared to HelioClim3, but our method provides others advantages.

Developing this new method aims at extending the cloud index concept to a broad range of satellite imagers of different sensitivities on different orbits. The way chosen

for that is to use radiative transfer simulations instead of archives of satellite imagery. This paper is a first step: we are able to produce DSSI estimates with a similar quality compared to operational products including HelioClim3 which is based on a cloud index method. The extension of the validation to other satellites including non-geostationary is an ongoing work that we aim to submit in the near future.

The new method also aims at investigating on the origin of cloud index uncertainties. Using simulations of TOA reflectances integrating surface, clear atmosphere, and cloud properties provides flexibility for future improvements and sensitivity analyses. We also consider that the development of an alternative method to compute the cloud index with different assumptions is useful to assess, for example, the robustness of DSSI time variations within multi-model comparison exercises. This work is exploratory, and our publication comes as a first version showing encouraging results. Several significant sources of errors are identified (source of calibration gains, spectral interpolation of MODIS BRDF data, cloud properties used in the $\rho_{ovc}$ Look-Up Table, angular description of the LUT). These errors will be further considered in future works (including on-going works), and their treatment is likely to improve results. Please also note that, as the paper focuses on the computation of the cloud index, the clear-sky index/cloud-index relationship is not investigated, and may also improve the quality of future results.

The aim of the method is clarified in the introduction (see answer to general comment 1.b.)

Moreover, why do they provide information on the correlation only?

We add in the abstract information on bias and RMS

4. Something is amiss in the logic of the approach: the Heliosat idea is to use a cloud index to get Downwelling surface solar irradiance (DSSI). This, for simplicity of the process and contrary to the LUP table approach that is based on simulations. In order to use the LUP tables one needs to know the parameters used in the simulations to do the matching with the observed TOA radiance/albedo. Not clear what is the benefit in doing the simulations that are not appropriately utilized?

The simulations are made to estimate reflectances as would be measured by a sensor at the top of the atmosphere in boundary cases clear-sky and overcast conditions. LUT are not used to estimate directly the DSSI. We use the radiative transfer simulations to compute a cloud index to estimate the attenuation of solar irradiance by clouds. Please also refer to the answer to the comment n°3 for more explanations on the objectives of the method.

5. The argument that the simulated SAL is better than the library of min SALs or that it can be used with every satellite, is weak. To estimate the DSSI for each case using the Heliosat approach one needs the SAL at the time of the observation. How is such matching achieved?

Our current version of the method deals only with historical time series of input data. A near real time version of the method could also be developed based on alternative datasets describing surfaces and clear atmosphere (e.g. climatologies, forecasts…)

Some misleading and unsubstantiated statements:

It is stated: "*the lower boundary is "archive-based", in most literature we reviewed: it is a minimum based on a time series of past satellite imagery. Such an approach is hardly applicable to non-geostationary satellites due to variable viewing geometries and a low*

*revisit time In this paper, we aim at finding an alternative to the need for archives of satellite imagery. It would then be easier to consider imagery from non-geostationary spaceborne platforms and produce a worldwide coverage.*

It was not shown how the simulated albedo is used in the context of geostationary satellites and/or polar orbiters.

We changed this part in the Introduction. To make it clearer, we mentioned the use of radiative transfer model to estimate the TOA clear-sky reflectance.

Stated:

*Heliosat-V is a method approximating the attenuation of DSSI radiation by clouds with a cloud index, n. We aim at developing an alternative "stateless" method to extend the application field of the cloud-index approach to a wider variety of orbits and optical shortwave sensors* What is "stateless"? How was it extended to polar orbiters? The paper deals only with SEVIRI. Briefly, in addition to the lack of clarity of the text it seems that it was not demonstrated that the stated objectives of improvement and generalization have been achieved.

We removed the expression "stateless". Our concept of simulating cloud index is able to be used on polar orbits, but as we explain in our answer to comment n°3, this paper is a first step: we are able to produce DSSI estimates with a similar quality compared to operational products including HelioClim3 which is based on a cloud index method. The extension of the validation to other satellites including non-geostationary is an ongoing work that we aim to submit in the near future.

In the section between lines 245-250 the following statements are made:

1. *The use of optimal calibration is out of the scope of our work. Still, we compared gains coefficients proposed by EUMETSAT gEUM with those provided by Doelling et al. (2018) gD2018 for the measurements produced by the Meteosat-9 250 0.6 and 0.8 μm channels in 2011.*

2. *They show a mean disagreement, calculated as (gEUM − gD2018)/gD2018, of about -9 % for 0.6 μm and -8 % for 0.8 μm during this period (also illustrated on Fig. A1). Such errors will affect with the same magnitude the agreement between numerical simulations and measurements of clear-sky TOA reflectances, underlining the importance of absolute calibration for the Heliosat-V method.* Not obvious what is the message of the Authors here: on one hand, the calibration is out of the scope of their work. Then they report on the evaluation of different gains which show large differences ( -9 %).

The identification of optimal calibration is not our purpose, but we emphasize the fact that different sources of calibration are different enough to cause errors on the cloud index computation depending on the calibration used.

They continue to state: *Such errors will affect with the same magnitude the agreement between numerical simulations and measurements of clear-sky TOA reflectance, underlining the importance of absolute calibration for the Heliosat-V method.* Which is it? Is it important or not?

We clarify by reformulating "This underlines that an accurate source of absolute calibration is important for the Heliosat-V method."

In Figure 6 provided are:

*Simulation of clear-sky reflectances at the TOA (ρclear) for MSG 0.6 μm (left panel) and 0.8 μm (right panel) spectral channels compared with actual satellite measurements. The comparison is done for all 11 locations, for the year 2011.* How was this comparison done? At each of the 11 locations, the atmospheric conditions are different. The atmospheric correction would be different. Not clear how the comparison was performed.

Figure 6 shows 2D histograms merging all clear-sky simulations and measurements. We clarify this point with "Represented data include simulations and measurements for all 11 locations, for the year 2011." We also add a description of statistics on STD for stations with best and worst results: "When studying station by station, the highest absolute standard deviation of the difference between simulations and measurements is reached for Sede Boker with 0.03, while the lowest is reached for Tamanrasset with 0.008."

In summary, this manuscript is not ready for publication.

---

## Author Response (AR2)

**Answer to Associate Editor's Comments to the author**:

The reviewers' comments seem to have been adressed satisfactorily in the responses. However, the changes in the revised manuscript (version 4) cannot be checked because the track-changes version is not provided. Please provide this.

We thank the associate editor for his comment, and provide here the track-changes version. (the AMT online system asks to provide a new version of the manuscript with the answer. Please note we provide the same document as in the previous submission of revised manuscript (version 4)).

---

## Author Response (AR3)

**File upload for the final accepted version**

The manuscript version uploaded here contains simple updated information for access to datasets (sections Code and Data availability), and completed acknowledgements.